# Temporal and spatial variability of Icelandic dust emissions and atmospheric transport

Christine D. Groot Zwaaftink[1], Ólafur Arnalds[2], Pavla Dagsson-Waldhauserova[2,3,4], Sabine Eckhardt[1], Joseph M. Prospero[5], Andreas Stohl[1]

[1] NILU Norwegian Institute for Air Research, Kjeller, Norway
[2] Agricultural University of Iceland, Hvanneyri, Iceland
[3] Faculty of Physical Sciences, University of Iceland, Reykjavik, Iceland
[4] Faculty of Environmental Sciences, Czech University of Life Sciences, Prague, Czech Republic
[5] Department of Atmospheric Sciences & Rosenstiel School of Marine and Atmospheric Science, University of Miami

*Correspondence to*: Christine Groot Zwaaftink (cgz@nilu.no)

**Abstract.** Icelandic dust sources are known to be highly active, yet there exist few model simulations of Icelandic dust that could be used to assess its impacts on the environment. We here present estimates of dust emission and transport in Iceland over 27 years (1990-2016) based on FLEXDUST & FLEXPART simulations and meteorological re-analysis data. Simulations for the year 2012 based on high-resolution operational meteorological analyses are used for model evaluation based on PM2.5 and PM10 observations in Iceland. For stations in Reykjavik, we find that the spring period is well predicted by the model, while dust events in late fall and early winter are overpredicted. Six years of dust concentrations observed at Stórhöfði (Heimaey) show that the model predicts concentrations in the same order of magnitude as observations and timing of modelled and observed dust peaks agrees well. Average annual dust emission is 4.3±0.8 Tg during the 27 years of simulation. Fifty percent of all dust from Iceland is on average emitted in just 25 days of the year, demonstrating the importance of a few strong events for annual total dust emissions. Annual dust emission as well as transport patterns correlate only weakly to the North Atlantic Oscillation. Deposition amounts in remote regions (Svalbard and Greenland) vary from year to year. Only limited dust amounts reach the upper Greenland Ice Sheet, but considerable dust amounts are deposited on Icelandic glaciers and can impact melt rates there. Approximately 34% of the annual dust emission is deposited in Iceland itself. Most dust (58%) however, is deposited in the ocean and may strongly influence marine ecosystems.

## 1 Introduction

Mineral dust is known to influence the radiation budgets of the atmosphere and cryosphere, ecosystems and human health. Even though fragile climate and ecosystems at high latitudes can be impacted, high-latitude dust sources have received rather little attention to date. Dust sources at high latitudes are often associated with glaciers. Glaciers produce fine material and, especially in floods, sand, silt and clay are deposited in glacio-fluvial plains from where they can be mobilized. Dust mobilization at high latitudes is strongly influenced by wind speeds, which are often quite strong in the presence of katabatic winds, sediment supply or dust availability, snow cover, freezing processes, and vegetation (e.g. Bullard et al., 2016). The

combination of these factors often leads to a strong seasonality in dust emission or dust storm frequency at high latitudes. High-latitude dust sources are for instance found at the coast in southern Alaska (Crusius et al., 2011), West-Greenland (Bullard and Austin, 2011) and Iceland (Arnalds et al., 2016).

It is known that dust storms frequently occur in Iceland. Analysis of weather observations showed that in the period 1949–
2011 on average 16 dust days occurred per year in north-east Iceland and 18 in south Iceland based on synoptic codes for dust observations (e.g. Dagsson-Waldhauserova et al., 2014a). In Iceland, not only dust from glacio-fluvial sources or sandur areas can be mobilized, but also tephra (material from volcanic eruptions) is re-suspended frequently and an important dust source (e.g. Arnalds et al., 2016). Dust storms in Iceland are not only frequent, but can transport large amounts of dust. For instance, a 24-hour mean concentration of particulate matter <10 μm (PM10) of 1281 μg m$^{-3}$ was recorded during a dust storm in
southern Iceland (Dagsson-Waldhauserova et al., 2015). Arnalds et al. (2013) reported average flux rates of 1440 kg m$^{-1}$ h$^{-1}$ over a 6.5-hour period in an erosion event of volcanic ash.

Impacts of such Icelandic dust storms are seen in air quality in Reykjavik (e.g. Thorsteinsson et al., 2011), glacier melt rates (e.g. Wittmann et al., 2017) and deposition of iron-rich material in the North Atlantic (e.g. Prospero et al., 2012) where it can fertilize the ocean (e.g. Achterberg et al., 2013). It is therefore important to know how much dust is transported to these regions
or systems. The studies mentioned here so far give valuable information on typical dust events in Iceland, yet partly lack quantitative information and do not consider long-range transport. Transport pathways from two main Icelandic dust source regions have been studied (Baddock et al., 2017) and qualitatively describe regions that may be affected. Dust emission amounts from Iceland were estimated by Arnalds et al. (2014). Based on storm frequencies, deposition rates, visibility observations and satellite images they concluded that 30.5 to 40.1 Tg dust is emitted annually in Iceland. Large uncertainties
in the extrapolation and conversion of visibility observations to concentration amounts (Dagsson-Waldhauserova et al., 2014a), however, limit the accuracy of this estimate.

Long-term model simulations could greatly improve dust emission estimates and help not only to identify regions possibly affected by Icelandic dust, but would also allow quantification of dust emissions and transport in regions where no measurement data are available. Global model simulations with FLEXDUST already indicated that 0.3% of global dust
emission may originate from Iceland (Groot Zwaaftink et al., 2016) during a three-year period, but temporal and spatial variability of Icelandic dust emission and transport were not discussed. Detailed modelling of Icelandic dust over a long period will help assess dust emission amounts and identify regions impacted by dust. Even for short events or periods, modelling of erosion is to our knowledge limited to studies of ash resuspension, for example of ash deposited during the Eyjafjallajökull eruption in 2010 and Grimsvotn eruption in 2011 (Leadbetter et al., 2012; Liu et al., 2014; Beckett et al., 2017). These studies
showed that timing of ash resuspension events could be represented with relatively simple models assuming fixed threshold friction velocities and accounting for the influence of precipitation. We here aim to model and discuss long-term dust emission with an adapted version of FLEXDUST (Groot Zwaaftink et al, 2016) and study dust transport with FLEXPART (Stohl et al., 2005). The complex interaction with the glacial system is currently not represented dynamically, but we use a highly detailed surface type map of Iceland (Arnalds, 2015) to identify dust sources. When referring to dust we here include volcanic material

that can be remobilized as well as mineral dust, although in our simulations we can only include the sources that are available from the surface type map. After introducing our model, we will present a brief model evaluation, discuss interannual variability of dust emission and transport, and estimate dust deposition to the ocean, Icelandic glaciers, Greenland and Svalbard.

## 2. Methods and data

### 2.1 Model descriptions

**FLEXDUST**

FLEXDUST, a model to estimate dust mobilization and emission, has been introduced by Groot Zwaaftink et al. (2016). This model estimates dust emission ($F$) as a function of friction velocity ($u_*$), threshold friction velocity ($u_{*t}$) and sandblasting efficiency ($\alpha$), based on the approach introduced by Marticorena and Bergametti (1995) and described by the following equation,

$$F = c\alpha \frac{\rho u_*^3}{g}\left(1 - \frac{u_{*t}^2}{u_*^2}\right)\left(1 + \frac{u_{*t}}{u_*}\right) \tag{1}$$

where $c$ is an added constant scaling factor set to 4.8*10^-4, consistent with global simulations presented by Groot Zwaaftink et al. (2016). The model is forced by analysis data of the European Centre for Medium-range Weather Forecasts (ECMWF). In global FLEXDUST simulations (Groot Zwaaftink et al.,2016) threshold friction velocities are based on sand fraction and a dependency on particle size according to Shao and Lu (2000), soil moisture influences threshold friction velocity according to Fécan et al. (1999), and sediment regions were identified based on large scale topography (Ginoux et al., 2001). For this study on Icelandic dust however, some adaptations were made.

For dust emission in Iceland, the model is combined with a surface type map presented by Arnalds (2015). As we have a highly detailed surface type map, we here do not include large scale topography effects to identify sediment regions in Iceland as was done by Groot Zwaaftink et al. (2016) to estimate global dust emissions. The surface type map is not changed throughout our model simulations, meaning that changes in dust sources due to for example volcanic eruptions are not accounted for.

The estimation of the threshold friction velocity for mobilization also differs from the standard approach in FLEXDUST. We use observations from Arnalds et al. (2001) and a description of erosion levels (Arnalds et al., 2016) to determine the threshold friction velocity (see Table 1). While Leadbetter et al. (2012) and Liu et al. (2014) chose a fixed threshold friction velocity of 0.4 m s^-1 for mobilization of volcanic ash, the range of values applied here is more suitable to cover the different conditions of multiple dust sources. Arnalds et al. (2016) give an overview of erosion classes for each surface type. For regions with extremely severe erosion we assume the average of threshold values observed at several sand fields, for severe erosion we assume average conditions of sandy gravel and for considerable erosion we apply an upper threshold observed for sandy gravel (Arnalds et al., 2001). So called dust hot spots, described by Arnalds et al. (2016), were also included in our simulations. These were assigned a lower friction velocity (see Table 1), corresponding to the lowest threshold wind velocity estimates for erosion

by Arnalds et al. (2016), and a slightly larger bare soil fraction (+3%). Bare soil fraction was assigned to dust sources based on surface type, varying between 0.65 and 0.95. A map of the Icelandic bare soil fraction in FLEXDUST is shown in Figure 1. In total, about $16.7 \cdot 10^3$ km$^2$ of the sandy deserts are categorised as active aeolian sources. Notice the close proximity of Icelandic dust sources to glaciers on Iceland, which is important for dust deposition on glacier surfaces. The combination of the field-based threshold friction velocity and the parameterization of soil moisture effects on threshold friction velocity (Fécan et al., 1999) normally used in FLEXDUST lead to low dust emission rates and modelled dust concentrations an order of magnitude lower than observed particulate matter concentrations at several stations in Iceland (see also section 2.3). It therefore appeared that soil moisture processes were wrongly represented by this combination of parameterizations and assumptions. Possible reasons for this are that threshold friction velocities obtained from Arnalds et al. (2001) were not observed during purely dry conditions, the parameterization by Fécan et al. (1999) is not applicable to the studied dust types or that soil moisture of Icelandic dust sources is not represented adequately in the meteorological analysis data we use. Thus, contrary to our previous work (Groot Zwaaftink et al., 2016), soil moisture does not affect threshold friction velocities in this version of FLEXDUST. Alternatively, we use precipitation as an indicator of decreased mobilization. In a model for resuspension of volcanic ash in Iceland, Leadbetter et al. (2012) assumed that precipitation can inhibit mobilization. Based on their model results, they concluded that a time lag before resuming mobilization after a precipitation event might improve model results. We tested the inclusion of such a time lag, but this did not improve simulation results (see section 3.1.1). Thus, in our current simulations, no dust emission occurs if precipitation exceeds 1 mm per hour and soil moisture has no influence on dust mobilization. The precipitation threshold is higher than the value of 0.1 mm/h used by Liu et al. (2014). In fact, they found discrepancies between model and observations that indicated that their threshold was set too low or that some time lag for the soil to become wet should be included.

We assume snow cover will inhibit dust emission if snow depth, retrieved from ECMWF analysis fields, exceeds 0.1 m water equivalent. In case dust sources near glaciers were falsely categorized as glaciers in the ECMWF data due to low resolution, snow depth at a reference point in interior Iceland was used. We further assume that the Westfjords area (west of 20°W and north of 65.2 °N) does not emit dust as it has a limited extent of dust sources (Arnalds, 2015). Indeed, in long-term observations, dust was found on only one day in five years in the Westfjords area, and this event could also have been caused by dust transport to the Westfjords from the central deserts (Dagsson-Waldhauserova et al., 2014a).

Emitted dust is assumed to have a size distribution according to Kok (2011), consistent with previous FLEXDUST simulations. Even though larger particle sizes have been observed in ash remobilization events (e.g. Liu et al., 2014), the Kok (2011) distribution appears more representative for the very fine material found in Icelandic dust sources and dust hot spots (e.g. Dagsson-Walhauserova et al., 2014b; Arnalds et al., 2016). Particles are split in 10 bins of different sizes; the first 5 bins are for particles from 0.2 up to 5 µm diameter, the remaining 5 bins extend up to 20 µm.

**FLEXPART**

FLEXPART 10.0 is used to calculate atmospheric transport of emitted dust from Iceland, and has previously been used to model the transport of Saharan dust (Sodemann et al., 2015) and globally emitted dust (Groot Zwaaftink et al., 2016). FLEXPART is a Lagrangian particle dispersion model (Stohl et al, 1998; 2005) driven by external meteorological fields. The model calculates trajectories of a large number of particles to describe transport and diffusion of tracers in the atmosphere. In FLEXPART, simulated dust particles are influenced by gravitational settling, dry deposition and in-cloud and below-cloud scavenging (Grythe et al., 2016). Dry deposition is treated using the resistance method (Stohl et al., 2005), wet deposition distinguishes between liquid-phase and ice-phase scavenging (Grythe et al., 2016). We used the default scavenging coefficients for dust and assume that particles are spherical.

## 2.2 Simulation setup

We did both high-resolution simulations for the year 2012 and a series of relatively low resolution simulations for the years 1990 to 2016. The high-resolution simulation in 2012 was based on hourly, 0.2° operational ECMWF analysis fields. The same analysis fields were used in FLEXDUST and FLEXPART simulations. Dust emission was calculated on a 0.01°resolution at hourly intervals with FLEXDUST. Emitted particles were gathered in hourly releases at 0.05° resolution. These releases were then used as input in FLEXPART simulations. The high resolution of dust emission fields allows us to benefit from the high-resolution surface type maps. Furthermore, initial particle locations are also more accurate, even though meteorological data and topography have a coarser resolution. Notice that this method takes advantage of the Lagrangian nature of FLEXPART which is, in principle, independent of the resolution of the meteorological fields and thus can ingest emission data at any resolution. The high-resolution simulation for 2012 included about 40 million particles.

The long-term simulations were based on 3-hourly ERA Interim reanalysis fields at 1° spatial resolution, in both FLEXDUST and FLEXPART. For these simulations, dust emissions in FLEXDUST were calculated at 0.02°-resolution on a 3-hourly basis and then gathered in 6-hourly releases at 0.5° for FLEXPART. For computational reasons the simulation was split into annual periods, each with an additional spin-up period of one month. Each annual simulation included on average roughly 10 million particles.

## 2.3 Observations

For model evaluation, measurements of concentration of particulate matter (PM) smaller than 10 µm (PM10) and smaller than 2.5 µm (PM2.5) are used together with dust concentrations. PM data are available at stations in Reykjavik (Grensasvegur and FHG), Hvaleyrarholt and Raufarfell, operated by the Environment Agency of Iceland. Locations are shown in Figure 1. The stations at Grensasvegur and FHG are equipped with a Thermo EMS Andersen FH 62 I-R instrument, the station at Hvaleyrarholt with Thermo SHARP model 5030 and the station at Raufarfell with Thermo 5014i. Observations were done hourly and averaged to daily values. PM measurements used here include PM10 and PM2.5, if available at the respective station, in the year 2012. In this year no volcanic eruptions occurred that could strongly influence PM measurements. Nevertheless, PM includes many particle types other than mineral dust (e.g. sea salt, anthropogenic emissions).

Dust concentrations were measured on Heimaey at a lighthouse at Stórhöfði (63°23.885'N 20°17.299'W, 118 m a.s.l.) on a daily basis with a high-volume filter aerosol sampler which collects total suspended particulates. Longer exposure times occurred occasionally due to bad weather and strong winds that precluded filter changing (Prospero et al., 2012). The observations were set up to study dust from remote sources, thus sampling was only done for wind directions south to west.

Measurements used here cover the period 8 February 1997 to 3 January 2003 and were averaged to weekly values.

## 3. Results and discussion

### 3.1 Evaluation

The possibilities for model evaluation are limited due to a lack of data in Iceland. Especially in north-east Iceland, where large dust sources are present, dust data are scarce. For earlier simulations using FLEXDUST and FLEXPART, Wittmann et al.

(2017) showed a comparison of modelled dust deposited on Vatnajökull and observed deposition in snow samples. They concluded that the modelled spatial distribution of dust deposition was similar to observations and dust deposition amounts were of the right order of magnitude. Satellite data are mostly valuable during strong dust events and require cloudless conditions and adequate overpass time of the satellite. Although visual inspection of MODIS images has confirmed particular dust events that will be discussed (such as in May 2012), they do not provide quantitative data and we do not include these.

Here, we restrict model evaluation to measurements of PM and dust concentrations in south-west Iceland.

### 3.1.1 PM concentrations

Concentrations of PM include different types of aerosols. Especially for stations near roads like Grensasvegur, concentrations are influenced by traffic emissions of PM. Nevertheless, dust storms are a recurring cause of episodes with elevated PM10 concentrations exceeding health limits ($>50$ µg m$^{-3}$) in Reykjavik (Thorsteinsson et al., 2011). About 1/3 to 2/3 of the days

with PM10 concentration exceeding the health limit in Reykjavik are likely caused by dust storms or by PM from local sources that may be dust as well (Thorsteinsson et al., 2011). Prospero et al. (1995) analysed aerosol samples taken at Stórhöfði in 1991-1993 for $NO^{3-}$, non sea-salt $SO_4^{2-}$ and methanosulfate and showed that concentrations thereof were similar to values measured in remote ocean regions for about 90% of the sample set. Peak values in 10% of the sample set were mostly related to aerosol transport from Europe. Moreover, observed nss-$SO_4^{2-}$ concentrations at Irafoss (Reykjavik) and Stórhöfði were

comparable during peak events.

The station Raufarfell,is located in the vicinity of dust sources and other influences are relatively small. Observed PM10 values (Figure 2) are frequently lower than PM2.5 values (Figure 3) in our data, even though this is, by definition, not possible. Since both quantities were measured with different instruments this can occur due to measurement errors in either of (or both of) the instruments. We have marked periods where PM2.5 values exceed PM10 values with grey shading in Figures 2 and 3. During

these days, observations either underestimate PM10 values or overestimate PM2.5 values, of which the latter is most likely given operational problems with these sensors.

In 2012 (Figure 2), several larger dust events occurred between May and November. There is a good agreement between the observations and the model at Raufarfell and most events are also represented in our FLEXPART simulation. In late September events are modelled at Raufarfell that were not visible in the observations, causing an overestimate of the number of days with concentration levels exceeding 50 µg m$^{-3}$ (Table 2). With the exception of the strongest dust event at the end of the measurement series, modelled concentrations are somewhat overestimating PM10 concentrations. This could also be related to topography, with the station placed in a mountain wind shade that might not be captured in the model. Nevertheless, the mean simulated concentration (28 µg m$^{-3}$) is close to the mean observed PM10 concentration (21 µg m$^{-3}$, Table 2), with almost identical standard deviations, indicating that dust variability is well captured. In Figure 2 we also show PM10 concentrations of a test simulation where we account for a time lag after precipitation in FLEXDUST. Here, we assumed that no dust emission will occur if the sum of precipitation over the last 4 hours exceeds 2 mm, since the sediments or soil need to dry before mobilization is possible. At this station relatively close to dust sources, it becomes clear that with such a time lag, several dust events seen in observations are no longer modelled and the default model is more representative. Probably, the material dries and can be remobilized relatively quickly, thus a drying period does not necessarily need to be accounted for. This is in agreement with observations of dust mobilization in Iceland during intermittent snowfall and wet conditions (Dagsson-Waldhauserova et al., 2014b, Dagsson-Waldhauserova et al., 2015).

All other measurement stations are located near or in Reykjavik and are further away from the dust sources, and closer to the ocean. This means that a) the measurements are less influenced by mineral dust and more strongly by other components (e.g. sea salt, road dust, pollution) and b) we expect larger discrepancies between model and observations because atmospheric transport and removal processes (and errors in simulating these) become increasingly important. At Hvaleyrarholt larger dust events, such as in May, are captured by the model. Differences between modelled and observed concentrations may of course also be influenced by the uncertainties in size estimates both in the observations and simulations, and in particular the effective size cut-off in the measurements. Especially during fall and early winter, PM10 concentrations are overestimated by the model. The results for PM2.5 (Figure 3) are very similar at this station. At the remaining stations in Reykjavik we clearly see increased background PM values (likely due to traffic). The model obviously underestimates these background values as only mineral dust is included in our simulations. Dust events are best recognized in peaks that occur simultaneously at FHG and Grensasvegur. Two distinct dust storms in May are indeed well represented by the model. The larger difference between measured and modelled PM2.5 than PM10 values may indicate that particle size distribution should be shifted, although it could also be due to a larger influence of anthropogenic aerosols on PM2.5 values. As for Hvaleyrarholt, we find that the number of dust storms reaching Reykjavik in fall and early winter is overestimated in the model output. Even though the dust storms at Raufarfell appeared nicely captured in this period (as far as measurements were available), it could be that other dust sources causing dust storms in Reykjavik are less well represented in our model. The highly dynamic nature of glacio-fluvial dust sources (e.g. Bullard, 2013) is not captured in our model and for instance depletion of specific dust sources during summer can explain the difference between model and observations. Furthermore, we apply a constant threshold friction velocity that

affects both timing and magnitude of modelled dust events. With source depletion and changing weather and soil conditions the threshold friction velocity might vary in time, causing a mismatch of model and observations in particular periods.

High PM10 concentrations in Reykjavik are a cause of concern. A health limit is set at 50 µg m$^{-3}$ and this should not be exceeded on more than 7 days per year (Thorsteinsson et al., 2011). In observations discussed by Thorsteinsson et al. (2011) this limit was reached up to 29 days per year. In 2012 the daily value of 50 µg m$^{-3}$ was exceeded on 7 days according to the measurements at Grensasvegur and on 16 days in the simulation (including only days with observations), as also shown in Table 2. The number of days with PM10 exceeding 50 µg m$^{-3}$ also appears overestimated at the other three stations (Table 2). Median values of modelled dust concentrations in Table 2, however, are generally lower than median values of observed PM10 concentrations, as expected since PM10 also includes other aerosol types.

Additionally, we compare weekly mean values of PM10 modelled at high resolution with ECMWF analysis data and at low resolution with ERA Interim data in 2012. The estimated emission in 2012 is 43% lower with ERA interim data (~2.9 Tg) than with hourly ECMWF operational data (~5.1 Tg). Because modelled dust emission has an approximate cubic dependency on friction velocity, higher time and space resolution – which better captures maxima in wind speed and thus friction velocity – can lead to higher emissions. Figure 4 shows that the modelled concentration values during dust events are not always decreased due to a lower resolution. Both episodes with higher and lower concentration values occur. Increases are for instance possible because dust emission grid cells can be larger and thus closer to the stations for the low-resolution simulations. This result thus shows that we cannot assume that a low resolution leads to generally lower concentration values. The results also show that modelled timing of events and order of magnitude of modelled concentrations are mostly maintained at low resolution. However, differences in model results cannot all be purely assigned to model resolution, as there are also other differences present between ERA Interim and ECMWF operational analysis data.

### 3.1.2 Stórhöfði - Heimaey dust concentration

The weather station at Stórhöfði is one of the weather stations in Iceland with the largest number of reported dust days in long-term records (Dagsson-Waldhauserova et al., 2014a). Stórhöfði is located on the Westman Islands 17 km off the south coast of Iceland (also see Figure 1) and a dust sampler has been operated here for many years (Prospero et al., 2012). In contrast to the PM measurements presented in section 3.1.1, the long-term measurements at Stórhöfði only include dust. Except for the period December 1999 – June 2000, the measurements were set up to measure mineral dust from remote regions (during winds from east through south to west) rather than Icelandic dust. Some local dust events may therefore not be recorded at all or underestimate actual dust concentrations, as only the fraction that 'returns' when the wind shifts to a direction within the sampling sector is included. The observations should thus be seen as a lower estimate of dust concentrations.

Weekly mean values of modelled and observed dust concentrations are compared over a period of approximately 6 years in Figures 5 and 6. The dust at Stórhöfði likely originates mainly from the coastal dust sources in south Iceland (see Figure 1). The mean values of observations and simulation during the complete measuring period are 8.9 µg m$^{-3}$ and 10.2 µg m$^{-3}$, respectively. The root mean squared error between model and observations is 17.6 µg m$^{-3}$. For the period when sampling was

not restricted to wind directions south through west, observed and modelled mean values are 12.7 µg m$^{-3}$ and 11.7 µg m$^{-3}$ respectively. We find that, except in 1999, the timing of peak dust concentrations appears to be very well captured by the model. This may be because these peaks represent large scale events rather than the activity of a few specific dust sources. Some events are modelled that do not occur in the measurements, but these appear to be limited in number compared to the results for fall events in Reykjavik. This suggests that the deviations in Reykjavik were restricted to specific dust sources. Possibly, threshold friction velocity assumptions for specific regions are not valid, the meteorological fields do not capture the actual conditions affecting dust mobilization, or transport modelling is inaccurate due to for example deposition schemes and model resolution. The peak events are mostly underestimated by the model. Some of these events are linked to glacial outburst floods (jökulhlaups) that can increase sediment supply, for instance in 1997 and 2000 (Prospero et al., 2012). Our model currently accounts only for a fixed but endless sediment supply, thus such temporary increases in sediment availability are not represented.

### 3.2 Dust emission

### 3.2.1 Spatial distribution

We show mean dust emissions calculated with FLEXDUST for the years 1990 through 2016 to understand which of the sandy deserts are the most important dust sources. The long-term averaged emission map (Figure 7) identifies important dust sources in NE Iceland and along the south coast and shows a large similarity with bare soil fraction (Figure 1). Differences between bare soil fraction and emission patterns can occur due to snow cover, precipitation, storm occurrence and threshold friction velocity. For example, (north-) west of Langjökull glacier, dust emission amounts are large according to FLEXDUST because there is less snow cover than in the interior highlands according to the ERA Interim data used in these simulations. In NE Iceland, on the other hand, snow cover can inhibit modelled dust emission during the winter season. At the south coast, precipitation has a larger influence on dust emission than snow cover.

In our model setup we accounted for dust hot spots that frequently emit dust and are assumed responsible for a large part of total dust emission in Iceland (Arnalds et al., 2016) by lowering the threshold friction velocity. In Figure 7, however, these dust spots are not recognizable as such. Their size is too small (in total approximately 400 km$^2$ of 16.7·10$^3$ km$^2$ active aeolian Icelandic sources) and dust emission in our simulations is not large enough that they could strongly influence the total annual dust emission in Iceland.

For dust emission, episodes of strong winds are very important. We therefore also infer on how many days per year dust sources are active. We look at dust hot spots Dyngjusandur and Landeyjasandur in particular, and at a sandy field (see e.g. Arnalds et al., 2016 for a description) about 50 km north of Dyngjusandur. Dyngjusandur was on average active on 302 days per year. On many days however, dust emission is only small, and 90% of total dust is therefore emitted in 145 days. Sporadic dust events account for the greatest fraction of emissions with 50% of dust emitted on only 37 days. This is particular for dust hot spots, characterised by soils with low threshold friction velocities. Further north of Dyngjusandur, in a sandy field some

dust emission occurs on 227 days, but 50% of dust is emitted in only 26 days. Similarly in the south, we find that the Landeyjasandur dust hot spot is active on 289 days, yet emissions on 38 days account for over 50% of annual dust emission. Looking at total dust emissions from Iceland, 50% is emitted in 25 days, and 90% in 110 days of the year. Previous studies of long-term dust frequency reported 135 dust days per year including minor events (Dagsson-Waldhauserova et al., 2014a). Given the dependency of this observation on the number and location of observations this is a good agreement. Days with largest dust emission occur in winter/early spring according to FLEXDUST.

### 3.2.2 Interannual variability

The average annual mean dust emission in the period 1990 until 2016 is 4.3±0.8 Tg. This is similar to the FLEXDUST estimate for dust emission in Iceland in years 2010 through 2012 in global simulations (4.8 Tg, Groot Zwaaftink et al. 2016). Dust emission rates are an order of magnitude lower than previous estimates of dust emission rates (30.5 to 40.1 Tg annually) presented by Arnalds et al. (2014). Their estimate includes dust spikes and redistribution in relation to volcanic events and glacial outbursts and is in part based on deposition rates (soil metadata and tephrochronology). Also larger particles are included in estimates of Arnalds et al. (2014), most of which would be deposited in the near vicinity of their sources. Other possible causes for this large difference are the large uncertainty related to extrapolation of visibility and storm frequency observations to dust concentration and emission estimates. Such estimates are also highly dependent on observation locations. An under-estimation of dust activity from the localized hotspots in our estimate can also not be ruled out. Nevertheless, such high emissions as reported by Arnalds et al. (2014) would lead to strong overestimates of observed concentrations with our model, unless the extra mass would be attributed almost exclusively to larger particles that never reach the measurement stations.

The North Atlantic Oscillation (NAO) is an important mode of meteorological variability in the North Atlantic and Europe (Hurrell et al., 2013). According to Polar MM5 simulations by Bromwich et al. (2005), changes in the NAO modulation of regional climate influence precipitation patterns in Iceland through shifts in the Icelandic low. To analyse whether the NAO also influences dust emission in Iceland we plotted time series of annual dust emission and the annual station-based NAO index (retrieved from Hurrell & National Center for Atmospheric Research Staff, 2017) in Figure 8. With a coefficient of determination ($r^2$) between annual dust emission and annual NAO index of 0.13 we find only a weak correlation. Distinguishing between dust emission from sources in south Iceland (<64.3 °N) and north Iceland (see Figure 8, right panel) shows that dust emission in south Iceland more strongly correlates with NAO index ($r^2 = 0.23$) than emission in north Iceland ($r^2 = 0.10$). The lack of a substantial correlation between dust emission and NAO is consistent with conclusions of Dagsson-Waldhauserova (2013; 2014) based on dust storm observations that the main driver of dust events is probably a pattern orthogonal to NAO.

### 3.3 Aeolian transport and dust deposition

To understand the transport of pathways of dust from Iceland, we look at maps of mean dust load in the atmosphere and deposition on the surface. As expected, dust loads are largest close to the sources (Figure 9), as large fractions of the emitted dust are deposited after only short travel distances (Figure 10). Dust concentrations rapidly decrease with altitude; 40% of suspended dust is on average situated at altitudes below 1000 m above ground level and only 6 % at altitudes above 5000 m (not shown). This is consistent with the discussion on altitude distribution of high-latitude dust presented in Groot Zwaaftink et al. (2016).

Patterns of dust load and dust deposition are naturally very similar. Since emission estimates were an order of magnitude smaller than estimates of Arnalds et al. (2014), deposition estimates are as well, but distribution patterns are similar. We also estimate especially large deposition rates in the Atlantic Ocean north-east and south of Iceland. Because dust emission is larger in northern Iceland (see Figure 8) and the main wind direction during dust storms in north east Iceland is from the south (Dagsson-Waldhauserova et al., 2014a), the majority of dust appears to be transported northwards. But also dust deposition south of Iceland appears considerable. The mean dust load and deposition patterns are consistent with a recent study of Baddock et al. (2017) showing three-day particle trajectories of dust storms from a location in north-east and south Iceland, calculated with HYSPLIT (Draxler and Hess, 1998) between 1992 and 2012.

To further understand what drives dust transport patterns, we look into correlations of monthly time series of dust emission, dust deposition and NAO index. In Figure 11a correlation between annual dust emission and annual deposition at each point is shown. Naturally, correlations are high close to dust sources where many large particles will be deposited. Away from sources the dust plumes spread and correlations become smaller. We find that especially in the region north-north-east of Iceland correlations are large. This may indicate that transport patterns do not diverge significantly here, only dust amounts. Given this large correlation, we have normalized dust deposition to annual dust emission for further analyses in Figure 11b and 11c. Correlations between dust emission in north-east Iceland and normalized deposition (Figure 11b) show a similar (yet weaker) pattern as Figure 11a. Focussing on dust emission in south Iceland (Figure 11c), we find that correlations are generally weaker. The direction of dust plumes originating from these sources may be generally southwards, but probably varies from south-west to south-east. Even though we find some relatively large correlations between dust deposition north-north-east of Iceland and dust emission in south Iceland, we do not think that these are strongly linked but are rather caused by dust emissions in the north co-occurring with emissions in the south. The strong correlation between dust emission in north and south Iceland ($r^2$=0.67, also see Figure 8) means that we cannot properly separate influences of these two source regions on dust deposition in specific regions. Baddock et al. (2017) studied the trajectories from sources in both the south and north of Iceland separately and showed that dust from south Iceland was mainly transported southwards. Finally, even though we know that dust emission and NAO are not closely related (section 3.2.2), we investigate if dust deposition and NAO are, as transport pathways might be influenced by NAO. Transport of air pollution from Europe to the Arctic for instance is strongly linked to NAO (Eckhardt et al., 2003). However, Figure 11d shows that Icelandic dust deposition patterns correlate poorly with NAO.

## 3.4 Dust inputs to the ocean, glaciers and other regions

Dust occurrence affects marine and terrestrial ecosystems and the atmosphere and surface radiation balance. We therefore quantify the annual variability of Icelandic dust inputs to glaciers, the ocean and dust deposition in Greenland, Svalbard and Europe based on our model simulations. A large fraction of emitted dust does not travel far and is deposited in Iceland. This fraction is $1.5 \pm 0.3$ Tg (Figure 12) or 34 % of annual emission. The consequences of such dust deposition in Iceland are very dependent on what type of surface is covered by the dust. For instance, correlations between dust deposition patterns and bird abundance are shown by Gunnarsson et al. (2015) and impacts of dust on Vatnajökull albedo and melt rates were discussed by Wittmann et al. (2017). We estimate that a considerable amount of dust is deposited on Icelandic glaciers (approximately 0.2 Tg(~5%) or on average 16 g m$^{-2}$). With glacier retreat and thinning, both horizontal and vertical distances of glacier areas to dust sources become smaller, causing enhanced dust deposition over the remaining glacier areas, as for instance also observed in a Holocene record of the Penny Ice Cap (Zdanowicz et al., 2000). This constitutes an important climate feedback mechanism. Figure 10 shows that interannual variability of dust deposition on Icelandic glaciers is similar to that of deposition in Iceland as a whole.

According to our simulations, most of the dust emitted in Iceland is deposited in the ocean. Simulated dust deposition to the ocean was on average 2.5 Tg or 58% of annually emitted dust. This estimate is much lower than the 14 Tg estimated by Arnalds et al. (2014), consistent with lower FLEXDUST emission rates. Smaller fractions of emitted dust ended up in Greenland (2%) and Svalbard (<0.1%). Annual variability of dust deposited to the ocean closely follows dust emission. Annual dust deposition of Icelandic dust in Greenland is more variable. Probably conditions during single, particularly strong dust episodes have a large influence on dust deposition in Greenland. The same is true for deposition in Svalbard, where deposition amounts strongly varied in the first years of our simulation period. From Figure 10 one can also infer that dust deposition amounts in Greenland are highly variable in space. Annual Icelandic dust deposition amounts at the Greenland east coast occasionally reach values up to 1 g m$^{-2}$ yr$^{-1}$. On average however, dust deposition in Greenland is only about 0.04 g m$^{-2}$. Especially in north-west Greenland, Icelandic dust deposition amounts are low, with for instance mean deposition amounts of less than $5 \cdot 10^{-3}$ g m$^{-2}$ yr$^{-1}$ at NEEM Camp (77.45°N, 51.06°W). Most Icelandic dust stays in the near Arctic (>60°N), where on average about 78% of dust is deposited. However, only about 7% of emitted dust is deposited in the high Arctic (>80°N) in the years simulated in this study. The model confirmed that substantial amounts of Icelandic dust are deposited in the Arctic cryosphere and can influence surface albedo and melt in Iceland, Greenland and in other parts of the Arctic, as also suggested by Meinander et al. (2016). Their hypothesis is that Icelandic dust may have a comparable or even larger effect on the cryosphere than soot (Bond et al. 2013).

## 4. Conclusions

In this study we studied dust emission and transport from Iceland over a period of more than two decades through model simulations. The FLEXDUST emission model was slightly adapted for these simulations, such as through the inclusion of dust hot spots and the use of precipitation data to limit dust mobilization.

Simulations show that annual dust emission in Iceland is $4.3\pm0.8$ Tg on average in the years 1990 through 2016. These estimates are lower than values reported in the literature (e.g. Arnalds et al., 2014). Nonetheless, estimated dust emissions for the Icelandic sandy deserts (covering 22.000km², Arnalds et al., 2016) are approximately 0.2 kg m$^{-2}$ yr$^{-1}$ and are comparable to estimated dust emissions in the western Sahara (0.1 kg m$^{-2}$ yr$^{-1}$, based on Laurent et al., 2008). Moreover, annual Icelandic dust emissions account for ~0.3 % of global dust emission (Groot Zwaaftink et al., 2016). Annual variability of dust emission

in Iceland showed a weak correlation ($r^2 = 0.13$) with NAO index.

Transport model evaluation is based on dust and PM concentration measurements, even though the number of measurement stations in Iceland is very limited. It is thus hard to fully constrain dust emission estimates. We found better agreements between modelled and observed PM concentrations close to dust sources than far away from dust sources. This indicates that the dust emission model works well, at least for the sources contributing mostly to those measurements. In Reykjavik, we

found that model simulations perform well in spring, but include too many dust episodes in late fall and early winter, compared to PM10 observations. This may be related to the dynamic behaviour of glacio-fluvial dust sources, which include areas where sediment availability is dependent on glacial floods. This complexity is typical for high-latitude dust sources (e.g. Bullard, 2013; Crusius et al., 2011), but currently not captured by FLEXDUST. Also other dust sources may be depleted or get covered, for instance by lava, and require adjustment of the surface type map currently not implemented. Furthermore, assumptions on

the threshold friction velocity influence timing and magnitude of modelled dust events and may be less representative in specific periods as threshold friction velocity changes with surface conditions. Additionally, model evaluation based on PM observations is complicated by the inclusion of aerosol types other than dust, especially in domestic areas and near the coast. At Stórhöfði, near the south coast of Iceland, the timing of the peaks in dust concentration in our simulations compared well with the observed peaks in measured dust concentrations between 1997 and 2002. This suggests that the model is equipped to

predict especially the large scale dust events.

In north Iceland dust transport patterns appear persistent and directed north-eastwards, in south Iceland they are more variable. Emitted dust can travel over long distances, reaching Europe (3% of emitted dust) or Svalbard (0.1%). A large fraction of emitted dust, especially large particles, is deposited close to dust sources and therefore stays in Iceland (34%). Dust deposition on Icelandic glaciers is thus substantial, annually about 16 g m$^{-2}$, although this value is dependent on model resolution, due to

the close proximity of dust sources and glaciers.. Spatial variability of dust deposition on glaciers is large and dust is mostly deposited near glacier boundaries at low altitudes (also see Wittmann et al., 2017; Dragosics et al., 2016). Glacier retreat and thinning may thus be coupled to both an increase of dust source areas and decrease of the average distance of the glacier

surface to dust sources, meaning a positive feedback between the dust cycle and melt rates. Similarly, annually about 2% of Icelandic dust is deposited in Greenland, mostly at lower elevations.

Marine ecosystems and the carbon cycle may also be strongly affected by Icelandic dust. Most dust emitted from Iceland (58%) is deposited in the ocean, according to our simulations. Especially in regions north-north-east and south of Iceland deposition amounts appear considerable.

Our simulations indicate that most dust emission occurs in north-east Iceland. Unfortunately, this region is not covered well with observations and model verification is lacking. Future research should therefore also focus on these areas to improve descriptions of the dust cycle in Iceland and quantify impacts on the climate system. Further research is also needed to better understand the dynamic changes in dust source regions due to volcanic eruptions. Re-suspension of volcanic ash is currently often treated separately from dust mobilization (e.g. Leadbetter et al., 2012; Liu et al., 2014; Beckett et al., 2017), although both processes are closely related and treatment of these sources should be unified.

**Acknowledgements**

We thank Thorsteinn Johannsson (Environment Agency of Iceland) for providing the PM observations and discussions. We acknowledge funding provided by the Swiss National Science Foundation (grant 155294) and travel grants provided by the Nordic Centre of Excellence eSTICC (Nordforsk 57001). OA and PDW were supported by Icelandic Research Fund (Rannis) Grant No 152248-051 and PDW by The Recruitment Fund of the University of Iceland. The station at Storhofdi was initially established with support from the US National Atmospheric and Oceanic Administration to JMP and later sampling and analysis with support various grants from the US National Science Foundation (AGS-0962256). We thank F. Beckett and two anonymous reviewers for their insightful comments.

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

**Table 1 Threshold friction velocity based on observations presented by Arnalds et al. (2001) in each erosion class described by Arnalds et al. (2016).**

| Erosion class | Threshold friction velocity (m/s) |
|---|---|
| Dust hot spot | 0.27 |
| Extremely severe (5) | 0.33 |
| Severe (4) | 0.58 |
| Considerable (3) | 0.70 |

5 **Table 2 Statistics on observed PM10 concentrations (µg m$^{-3}$) and simulated dust (d<10 µm) concentrations (µg m$^{-3}$) at four stations in Iceland.**

| | Raufarfell | | Hvaleyrarholt | | Grensasvegur | | FHG | |
|---|---|---|---|---|---|---|---|---|
| | Obs. | Sim. | Obs. | Sim. | Obs. | Sim. | Obs. | Sim. |
| Median concentration | 9 | 4 | 6 | 2 | 11 | 2 | 10 | 2 |
| Mean concentration | 21 | 28 | 8 | 10 | 15 | 9 | 13 | 10 |
| Standard deviation of concentration | 95 | 89 | 9 | 17 | 14 | 17 | 11 | 18 |
| Number of days PM10 > 50 µg | 13 | 31 | 3 | 17 | 7 | 16 | 3 | 14 |

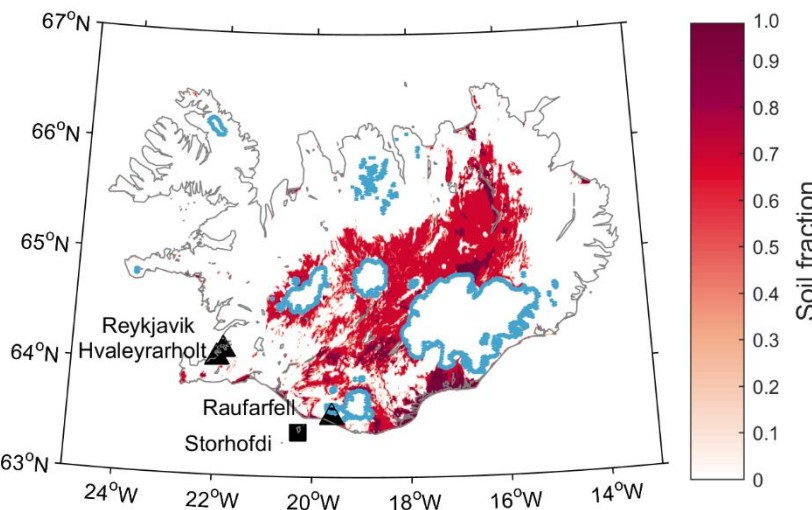

**Figure 1 Aeolian active bare soil fraction as assumed in FLEXDUST. The triangles indicate stations with PM measurements. The square marks the Storhofdi station with dust concentration measurements. The blue lines are glacier outlines.**

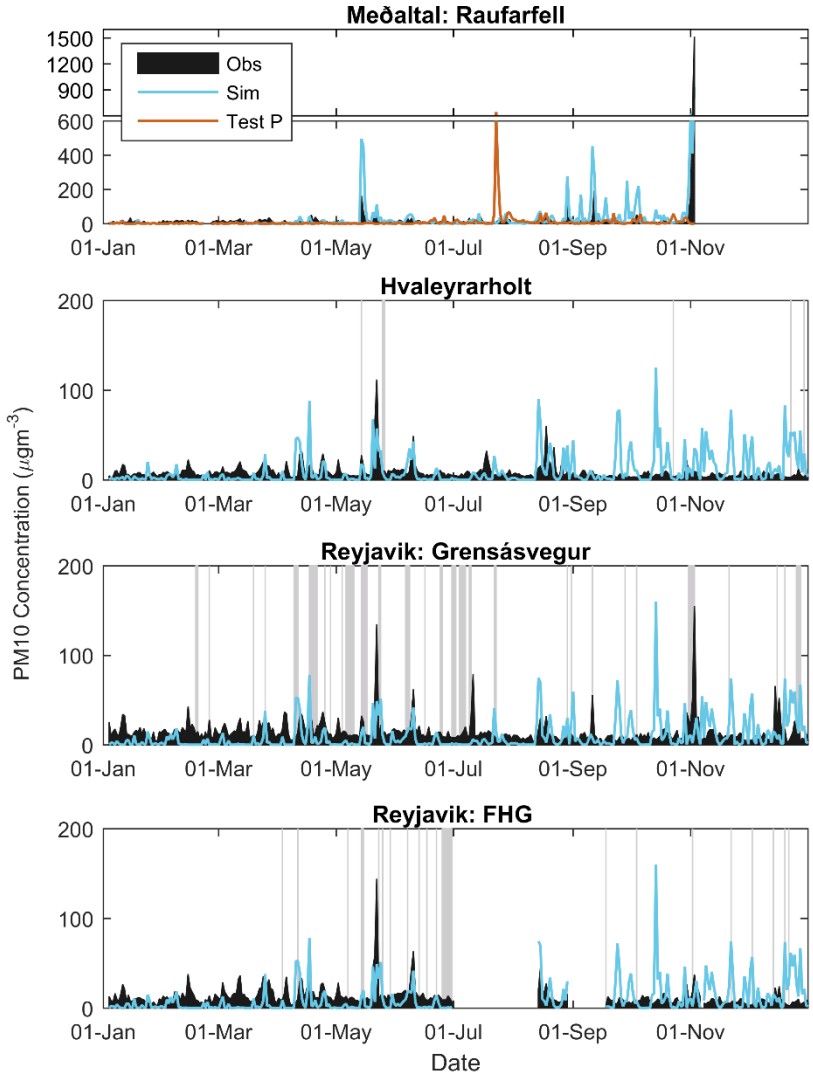

**Figure 2 Daily mean PM10 concentrations (µg m⁻³) as observed (black) and modelled (blue) in 2012. A simulation where a time lag after precipitation was taken into account is shown at Raufarfell (Test_P, orange). Shaded grey areas indicate periods with inconsistent measurements of PM10 and PM2.5 (also see figure 3).**

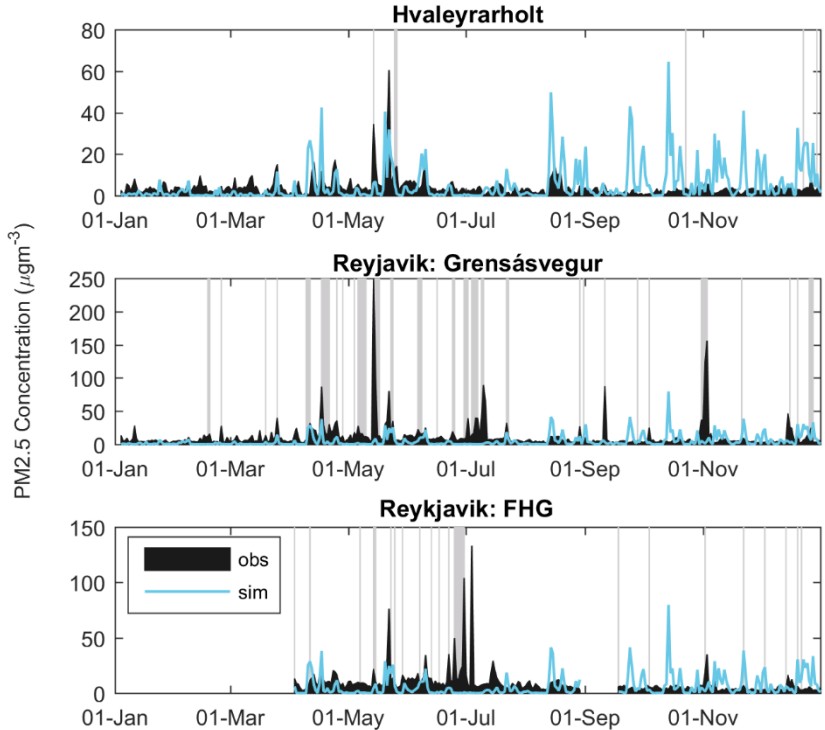

**Figure 3 Daily mean PM2.5 concentrations (μg m⁻³) as observed (black) and modelled (blue) in 2012. Shaded grey areas indicate periods with inconsistent measurements of PM10 and PM2.5 (also see figure 2).**

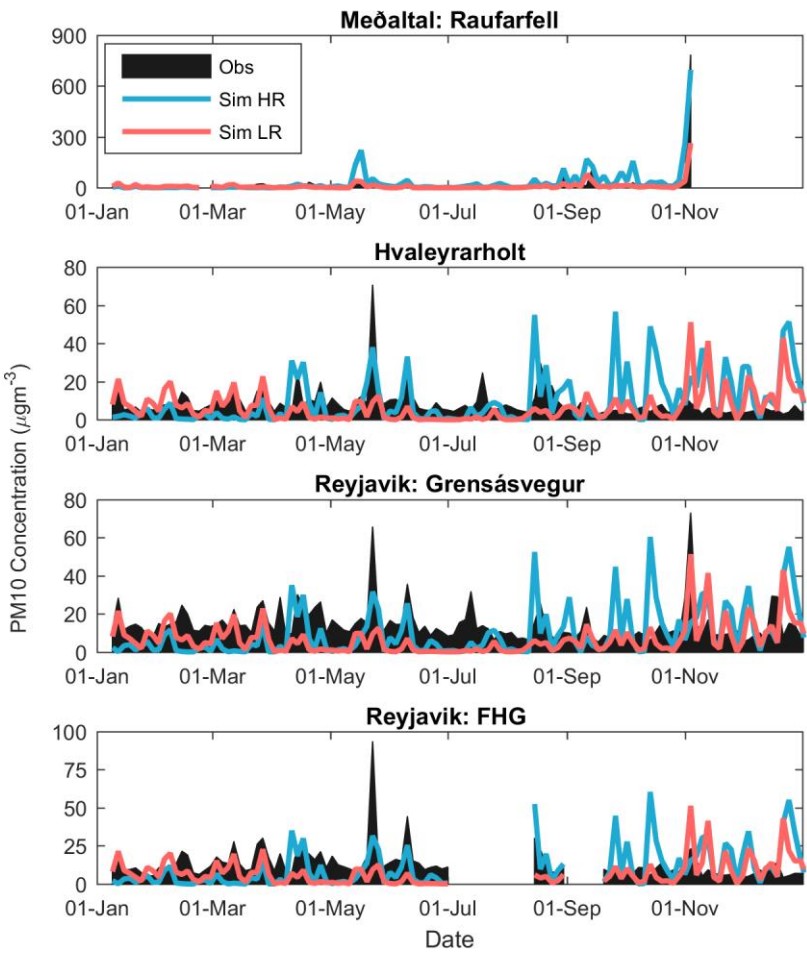

**Figure 4 Weekly mean PM10 concentrations at four stations as observed (black), modelled at high resolution (blue) with ECMWF analysis data (0.2°) and modelled at low resolution (1.0°) with ERA Interim data (red).**

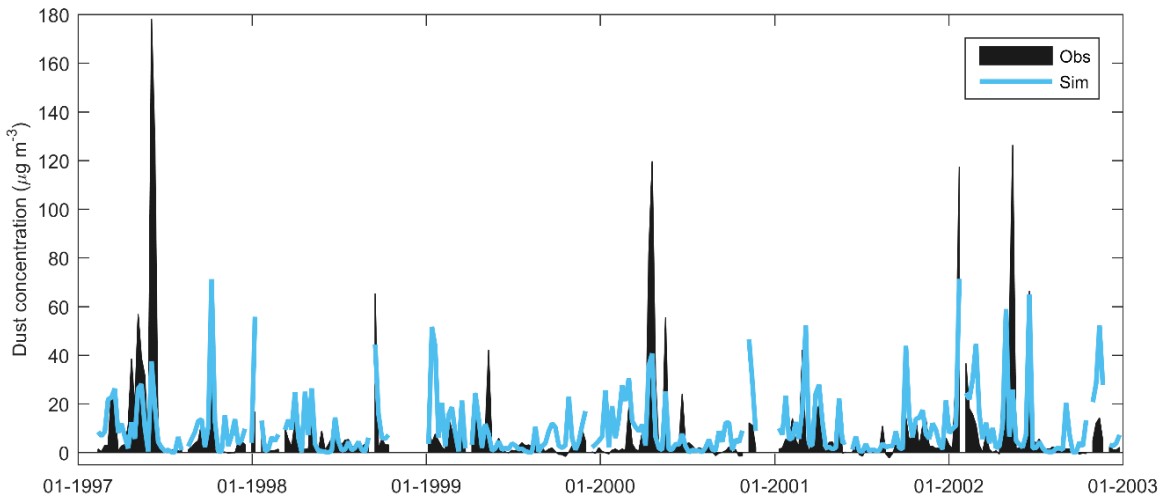

**Figure 5 Observed (black) and modelled (blue) weekly mean dust concentration (µg m⁻³) at Stórhöfði /Heimaey.**

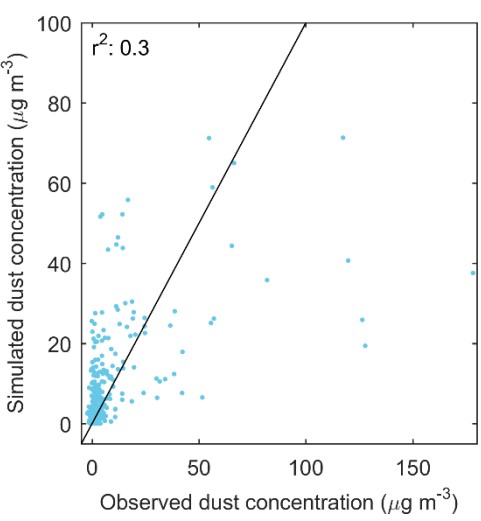

**Figure 6 Weekly mean simulated versus observed dust concentration (µg m⁻³) at Stórhöfði /Heimaey. The black line shows where simulated and observed values are identical.**

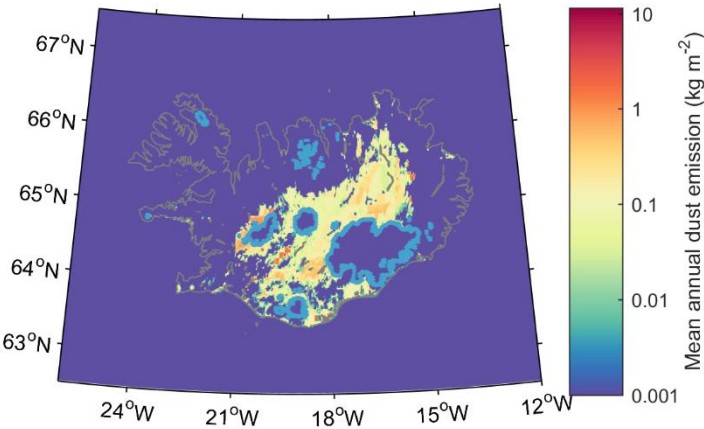

**Figure 7 Simulated annual mean dust emission (kg m⁻²) in years 1990-2016**

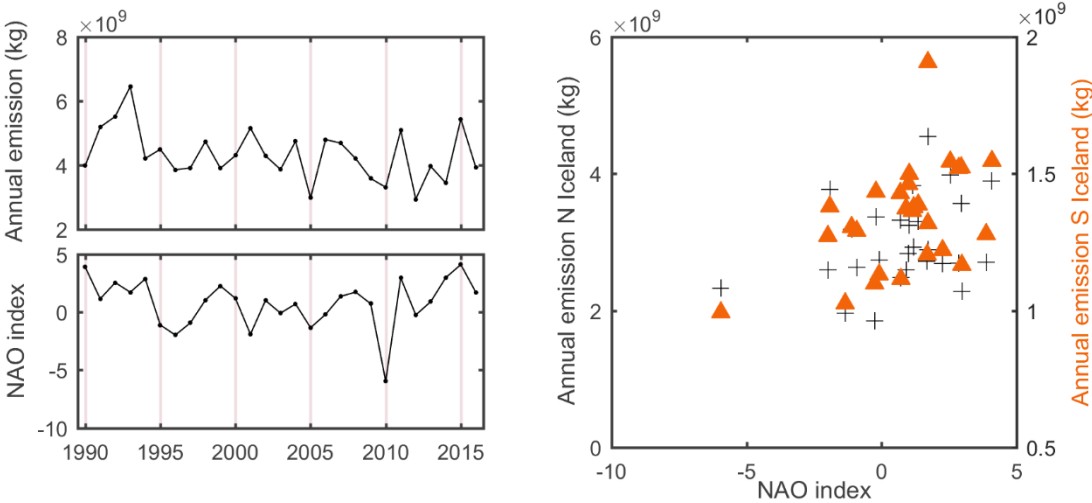

**Figure 8 Left: Annual dust emission from Iceland in years 1990 until 2016 (top) and the annual NAO index (bottom). Right: Annual emission from Northern Iceland (>64.3˚N) and southern Iceland (<64.3 °N) versus annual NAO index.**

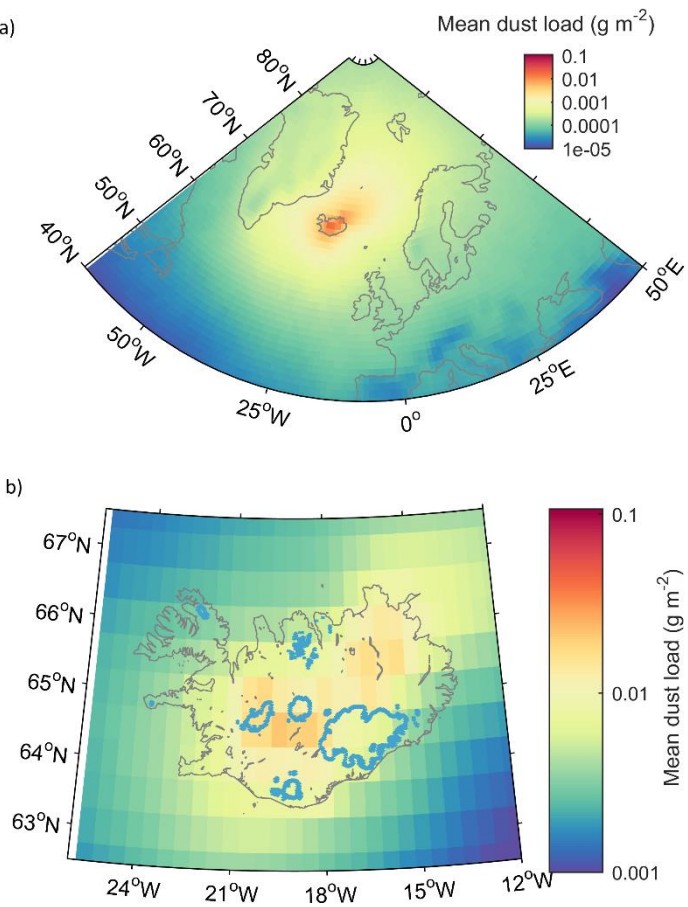

**Figure 9 Mean atmospheric dust load (g m⁻²) simulated with FLEXPART in years 1990-2016 for the North Atlantic region (top) and Iceland (bottom). The blue lines in the bottom figure are glacier outlines.**

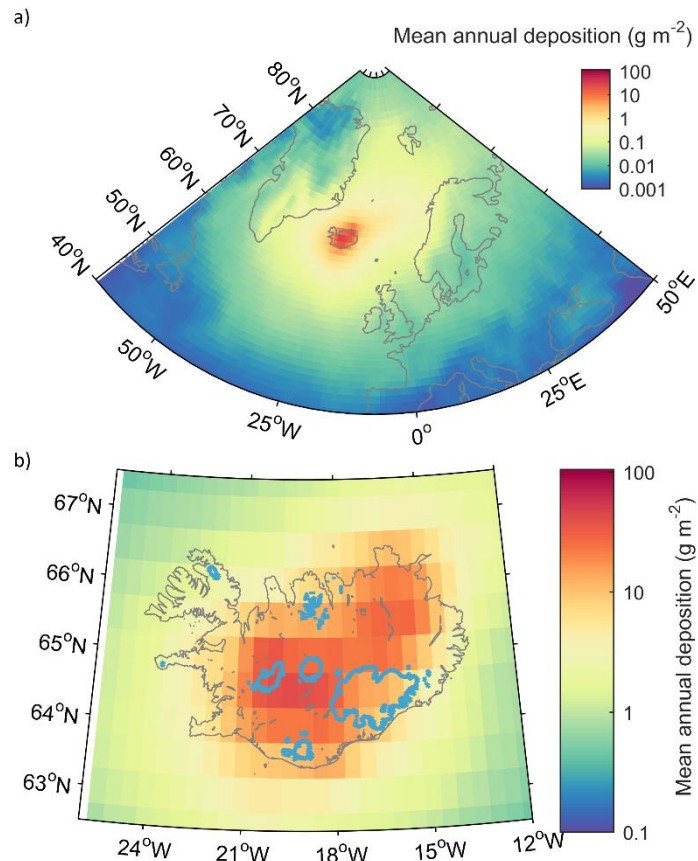

**Figure 10 Mean annual dust deposition (g m⁻²) simulated with FLEXPART in years 1990-2016 for the North Atlantic region (top) and Iceland (bottom). Maximum values are lower in the upper panel than in the lower panel as this figure shows averages over larger areas. The blue lines in the bottom figure are glacier outlines.**

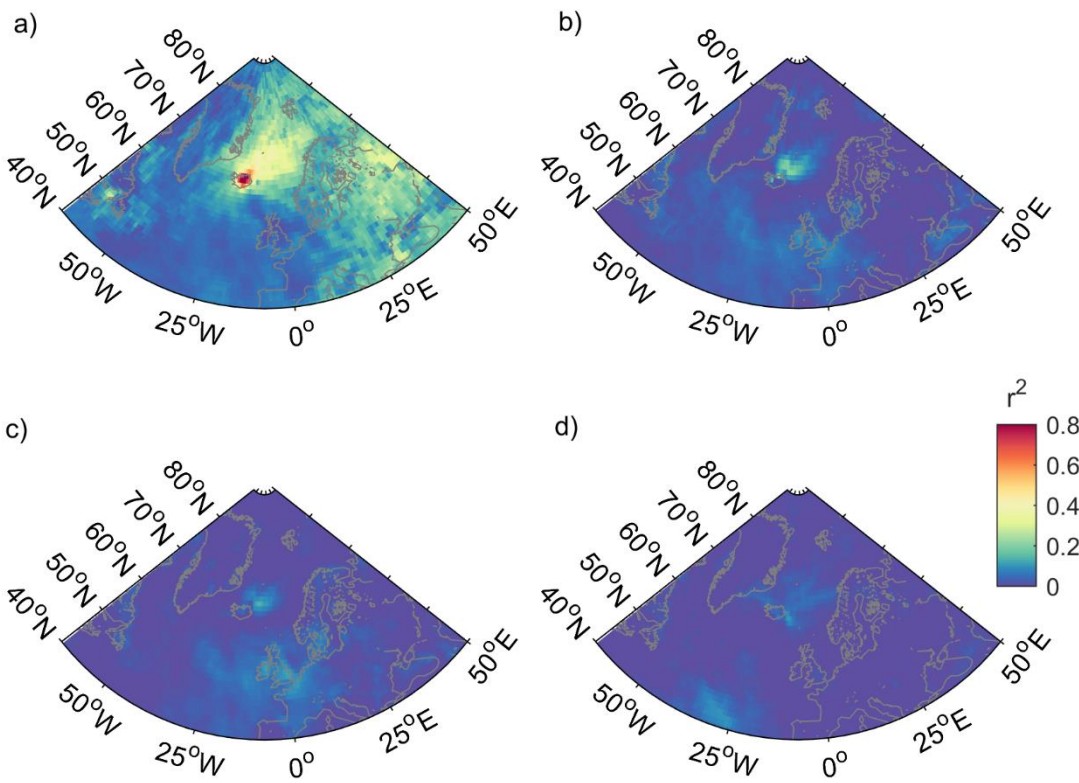

**Figure 11 Coefficient of determination r² for monthly time series of dust deposition and emission (a), dust deposition normalized by total emission and emission in N Iceland (b), dust deposition normalized by total emission and emission in S Iceland (c), dust deposition and the NAO index (d).**

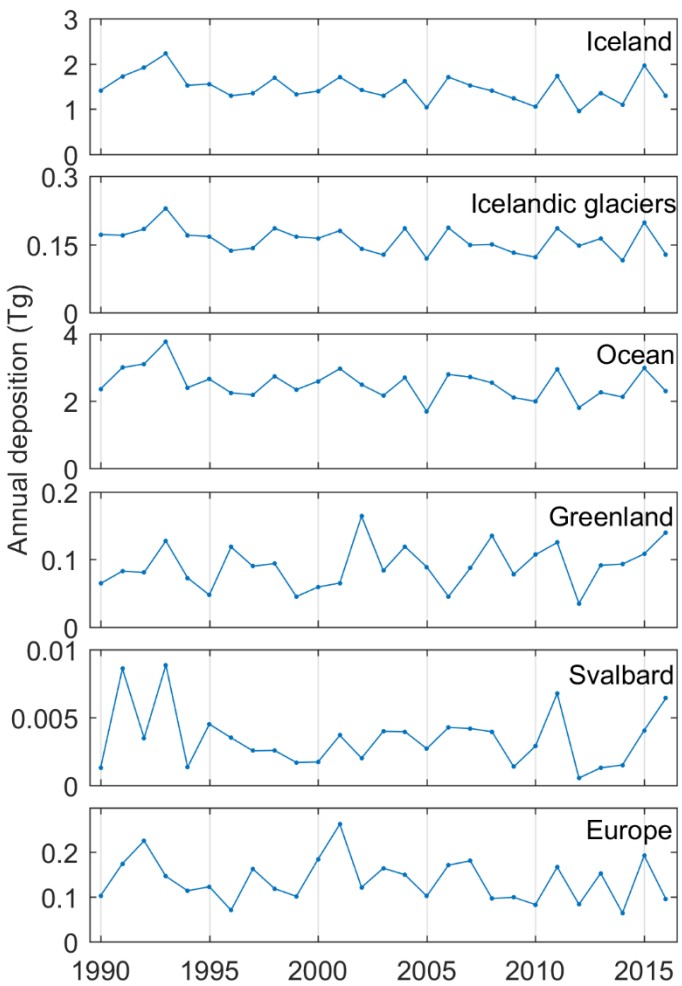

**Figure 12 Time series (1990-2016) of modelled dust deposition (Tg y$^{-1}$) in specific regions. Note that Iceland also includes deposition on Icelandic glaciers.**