# Peer review of "Temporal and spatial variability of Icelandic dust emissions and atmospheric transport"

_Atmospheric Chemistry and Physics, 2017_

## Referee Comment (RC1) · Anonymous Referee #1 · 10 May 2017

The authors present results from a muli-annual (27 years) study assessing the capability of the Lagrangian model FLEXPART to capture the Icelandic atmospheric dust life-cycle. Thereby, dust emission fluxes are estimated using FLEXDUST. Results of their study were further discussed regarding its interannual variability; results at high resolution were validated against measurements for the year 2012.

The manuscript is well structured and a nice read. However, I do have some comments I would like the authors to address.

2.1 Model description

(1) In the subsection FLEXDUST you describe how dust sources were implemented in the model. You state that lower friction velocities and large soil fractions were assigned to dust hot spots as identified by Arnalds et al. (2016). I am wondering whether these dust host spots occur due to enhanced levels of sediment supply or due to higher frequencies of stronger winds (maybe also channelled by orography).

(2) Can you spend some more words on how FLEXPART and FLEXDUST co-exist respectively intertwine as this remains somewhat diffuse. As far as I understand FLEXDUST is used to estimate dust emission fluxes based on ECMWF forecast analyses at 0.2deg horizontal grid spacing. The calculated emission fluxes are then read into FLEXPART and transported whereby FLEXPART is driven using the ERA-Interim reanalysis at 1deg horizontal grid spacing. Why were two different atmospheric data sets chosen to drive the models rather than using consistently ECMWF forecast analyses for both but on a different horizontal grid?

(3) How is dust deposition respectively removal parameterized? Please add some explaining words. Is wash-out and scavenging due to rain and clouds considered as particle removal processes?

(4) Simulation setup (section 2.2): As the input meteorological fields were available at a grid with a 0.2deg horizontal grid spacing, but dust emission fluxes were estimated on a grid with 0.01deg horizontal grid spacing, can you explain if there has been any upscaling or interpolation method applied, please? Is topography taken into account for the upscaling?

3. Results and discussion

(5) In section 3.2.1, numbers of days of active dust emission are provided as

fraction per annum. How do these numbers of days compare to seasons? Some additional sentences presenting and discussing the seasonal distribution of dust emission events, transport and deposition can help here to draw a more thorough picture of the Icelandic atmospheric dust life-cycle - and eventually imply further mechanism controlling interannual variability.

(6) Is there any explanation why the NAO has no significant correlation with dust emission in Iceland? (section 3.2.2)

(7) As stated in section 3.2.2, the NAO has no significant impact on dust emission. However, why is the NAO used as measure describing Aeolian transport and deposition patterns (section 3.3)? May topography has an important and maybe dominating impact on the transport direction here?

(8) How is the dust vertically distributed? Is there any significant dependency between dust deposition region and transport height or mixing depth into the boundary layer over source regions that can be concluded from the FLEXPART simulations? An enlarged discussion on dust transport pattern and deposition regions is desirable in order to clarify the conditions under which Icelandic dust is transported far beyond its source region. Furthermore, the results may vary with season as the predominance of meteorological situations (e.g. occurrence of precipitation, cloud formation) and atmospheric circulation patterns changes.

(9) Can the hypothesis by Meinander et al. (2016) that "Icelandic dust mays have a comparable or even larger effect on the cryosphere than soot" be confirmed by the presented study?

---

## Referee Comment (RC2) · Anonymous Referee #2 · 18 May 2017

This compact paper, "Temporal and spatial variability of Icelandic dust emission and atmospheric transport" presents surface observations and results of lagrangian simulations of dust emission and deposition at high resolution for 2012 and lower resolution for 1990-2016 to estimate the dust emission and deposition to the region. The paper is very well written, presents interesting results from modeling and the observations, and references the appropriate literature. I believe that details are lacking in places and the analysis is a little weak, namely the comparison with observations and the certainty with which the dust emissions can be estimated, and would like to see those parts improved prior to publication.

General comments

[Figure]

The PM10 and simulated dust concentration yield similar mean (21 ug/m3 and 28 ug/m3, respectively) and standard deviations (pg 5 line 10); however, this is comparing dust-only concentrations from the model with bulk aerosol PM10. This suggests that the simulated dust concentrations are actually biased high (maybe up to a factor 2?) relative to the observations (if the non-dust component could be removed from the PM10). I think the way that the model results are compared to the PM10 (and PM2.5) may need reconsidering or the present method better justified. Can you estimate the non-dust component? How much of the PM10 at the sites may be localized dust that would not be captured by the model?

This affects the attempt to estimate the annual emissions from Iceland and subsequent deposition. I'm not sure whether the current observational constraints and analysis are able to fully support the estimate. The agreement with the dust concentration measurements seems reasonable at StórhöfÃři, but this is only a single measurement site SW of the source regions. Therefore, the constraint on emissions transported in other directions is weak; it appears that equal, if not great, dust mass is deposited to the NE. Could the statistical relationship between observations and modeled dust concentration be used to better estimate the emission, or at least the uncertainty? For example, how much would the emissions need scaling to provide the same average dust concentration (or some other metric) at StórhöfÃři? this suffers from the lack of constraints for the dust in the NE, but might give a better representation of the dust emissions and their uncertainty beyond the interannual variability.

Following from this, I can't see any comparison of the low and high resolution runs in 2012 (other than 2.9 Tg and 5.1 Tg totals for 2012 on pg7, line 21). Does this mean that running at high resolution may give 75% higher emission estimates than the 4.3+/-0.8 Tg presented for the long term estimates? I don't think the implications of this are discussed clearly enough. The uncertainty estimate for the interannual variability may mislead the reader to the certainty of the magnitude of the dust emissions (and hence deposition).

[Figure]

Does the low resolution run well-reproduce the high resolution simulated dust concentration timingin 2012 otherwise? Maybe add the low resolution timeseries at StórhöfÃři to Figure 2?

While the time series of concentration provide a good visual reference of the frequency and magnitude of events, they are not ideal for illustrating the agreement between the simulation and the observations. I recommend providing a scatter-plot (perhaps on a log-log scale?) to better illustrate how well the model captures the observations of dust concentration. This is less useful (and therefore perhaps less necessary) for the comparison with PM for the reason outlined above, unless speciation is available.

Emissions are not allowed when the precipitation rate is above the 1 mm/hr threshold. Is there a lag time for this emission suppression after the rain stops? Or is it expected that the timescale for the surface drying and becoming an active once again is shorter than the model timestep? How much do you think this assumption affects the emission?

In Groot Zwaaftink et al. (2016) it is stated that, relative to a precipitation threshold, "Especially in northern latitudes, soil moisture appeared a better indicator of mobilization threshold as seasonal variations in surface dust concentrations at remote stations were better captured and total emission amounts were closer to other model estimates." Please can you comment on why this is different to the current research findings for Iceland.

Specific Comments

pg 3 line 18 - "FLEPXART" typo

pg 7 line 19 "FLEXUDST" typo

It may be clearer to refer to the "soil fraction" as the "bare soil fraction" throughout.

Table 1 - it isn't quite clear how these values are derived from the threshold friction velocities presented in Arnalds et al. (2001) and the discussion in Arnalds et al. (2016).

Please can you elaborate in the text on how these values are derived.

Figure 2 - The Raufarfell timeseries is hard to see because of the upper limit. Is it possible to use a discontinuity on the y-axis above ∼600 ug/m3 to better visualize the data at lower concentrations.

---

## Referee Comment (RC3) · F M Beckett (Referee) · 12 Jun 2017

The paper presents a modelling study of the emission and transport of dust in Iceland between 1990 and 2016. It highlights the significance of high latitude dust sources on the global dust budget, and the authors present interesting results showing the main transport pathways of dust from Iceland. However, I believe the description of the model set-up needs to be significantly improved before this paper can be published. Details, including a description of the resolution of the model topography used and the particle size distribution applied are missing, and there needs to be some discussion on how their results may be sensitive to their set-up. The manuscript would also be

improved by including some discussion on how the supply of new dust sources, related to volcanic eruptions in Iceland, might influence their results.

1. The Introduction

I can see the importance and relevance of this study but I don't think this is reflected in the introduction. Details are missing and statements are often not backed up with existing data and/or references are missing. Currently, it reads as a series of statements rather than explaining to the reader why the study is important, the approach, and how it fits in with the existing literature. You need to discuss in more detail the work that has previously been carried out to better understand dust emissions in Iceland, including work published by Olafur Arnalds and Pavla Dagsson-Waldhauserova, and you should consider work on dust events in other parts of the world too.

Further discussion on modelling dust emissions is also needed. You state that model simulations of dust emissions in Iceland are lacking but there is now a body of work on modelling remobilisation of volcanic ash in Iceland, see Leadbetter et al. (2012), Liu et al. (2014), Beckett et al. (2017), and further afield, for example Folch et al. (2013) and Mingari et al. (2017) who consider remobilization in Argentina. Given that volcanic ash is a significant source of PM in Iceland (indeed there is the question of what is dust and what is ash!!), and the modelling approaches for remobilized ash are very similar to the approach you have applied here you should discuss this.

Specific comments:

Line 3: You state that: 'Model simulations indicated that 0.3% of global dust emission may originate from Iceland (Groot Zwaaftink et al., 2016)'. More details are needed here, what model, what simulations were performed and with what aim? If this has already been done then where does the study you are about to present fit in?

Line 4: You state that 'it is known that dust storms frequently occur there [Iceland]' and cite Dagsson-Waldhauserova et al. (2014). It would be good to include some numbers

here e.g. how many dusty days, on average, occur in Iceland. This will help put your results into context later on too. I realise you comment on this later in the paper but this should be here in the Introduction.

Line 14: You need to provide a reference for the surface type map that you refer to.

**2. Model Set Up**

The explanation of your model set-up is missing many details. I think you should include the equations used in FLEXDUST to model the emission of dust, and explain the variables. Exactly how does your model set-up account for topography, snow cover and soil moisture?

You state that precipitation halts mobilization. You need to refer to the work of Leadbetter et al. (2012) here who also considered how best to represent the impact of precipitation on mobilization of volcanic ash in Iceland. Please can you also comment on how well you think this approach is working in respect to representing the timing and frequency of dust events? This is discussed by Leadbetter et al. (2012) and Liu et al. (2014) who both point out that this approach does not account for wetting and drying of volcanic ash deposits, do you think this is true of all dust sources?

Please provide the Particle Size Distribution (PSD) you used and explain your reasoning for this choice. Why did you choose to consider particles with diameter up to 20 um only? What is the minimum particle size you considered? The work of Liu et al. (2014) gives the PSD of ash particles that had been remobilized and deposited in Reykjavik during March 2013. They found that particles had a mode at 32-63 um. Have there been any measurements of the PSD of particles mobilised from the other dust sources in Iceland?

**3. Thresholds Friction Velocities.**

Please provide your reasoning for the threshold friction velocities that you apply. How were these values determined from the Arnalds et al. (2001) and Arnalds et al. (2016)
papers and how are the classes defined? Please also provide information on how these classes are distributed across Iceland. Figure 1 shows the soil fractions applied but please also highlight where the Dust Hot Spots are and how the erosion classes are applied across the other source regions.

Please can you also comment on how good a job you think these threshold friction velocities are doing. By applying this range of values are you doing a good job of representing the timing and frequency of events in your model output? How sensitive is your model output to the threshold friction velocity applied? Can you account for some of the mismatch between the observed and modelled PM10 and PM2.5 air concentrations if you vary the threshold friction velocity applied?

4. Topography

What is the resolution of your model topography? Are your results sensitive this? You state in the introduction that dust events can be driven by katabatic winds; does your model topography allow you to capture these meteorological phenomena? Mingari et al. (2017) show how the topography in Argentina influences the local winds and in turn how that drives mobilization. I think you need to consider this. This information would help put in context your later comment in Section 3.1.1 that the model output may not be able to capture observed PM10 concentrations because of the resolution of the topography.

5. Sources

You compare your model output air concentrations to PM data from monitoring stations across Iceland collected during 2012. You state that: 'In this year no volcanic eruptions occurred that could strongly influence PM measurements' (Line 8, Section 2.3). I disagree. Do you really think that the ash deposits from the eruption of Grimsvotn only the year before and from Eyjafjallajökull in 2010 had all been removed and were no longer a significant source of PM? The study by Leadbetter et al. (2012) considers the remobilization of volcanic ash from the deposits resulting from the eruption of

Eyjafjallajökull in 2010. They compared modelled air concentrations using the dispersion model NAME, which includes a resuspension scheme, to PM10 measurements across Iceland during September 2010 to February 2011. Their modelled concentrations agree well with the timing and location of observed peaks in the PM10 data from the monitoring stations, and here only the Eyjafjallajökull ash is defined as the source. I recognize that your study aims to consider the long-range trends of dust emission and transport from sources across Iceland, but I think you need to acknowledge the fact that volcanic eruptions result in significant new sources of unconsolidated deposits which can continue to be remobilized for years after an eruption. In Section 3.2.2 you go on to state that your modelled dust emission rates are an order of magnitude lower than previous estimates given by Arnalds et al. (2014), and you say this could be related to volcanic events. I would suggest that you could explore this further and consider that the deposits from the Grimsvotn and Eyjafjallajökull eruptions could be a significant source of PM in your study.

6. The impact of NAO

I did not follow why you chose to consider the role of NAO as part of your study and what the significance is? What meteorological variables and/or synoptic conditions related to NAO do you think impact mobilization events in Iceland?

Minor Comments

In several places, including in the abstract, you state your conclusion that: 'Annual dust emission amounts to 4.3±0.8 Tg during the 27 years of simulation'. I find the term 'amounts to' a little confusing when discussing the yearly average. Please clarify.

Page 1, Line 3. Emission should read emissions.

Page 1, Line 19. 'A model for estimates of dust emission', does not read very well. The structure of this sentence needs to be improved.

Page 1, Line 26. Please provide references for your examples on the impacts of dust.

Page 2, Line 14. '.......surface type map of Iceland to identify dust sources'. I think you need to cite Arnalds et al. (2016) here.

Page 2, Line 21. I did not quite follow this sentence: ' ....and originally accounts for snow cover, topography....' . What do you mean by 'originally'?

Page 3, Line 3. 'As we here mainly deal with sediments'. What do you mean by this statement, what is the relevance of 'sediments' is this different to 'dust'. Please clarify. Also the structure of this sentence could be better, what do you mean by 'mainly deal with'?

Page 3, Line 8. What do you mean by a 'closed snow cover'?

Page 3, Line 17. 'as was previously also done for'. Could read better, how about 'and has previously been used to model the transport of Saharan dust'.

Page 3, Line 21. What do you mean by a 'multitude of particles'? Please be specific.

Page 4, Lines 1 and 2. Here you write the units of the particle size (micrometre), in other places you use the symbol. Please correct. Also, the structure of this sentence could be improved.

Page 4, Line 17. 'Model evaluation is limited due to a lack of data.' This sentence does not read well. Please improve the structure of this paragraph.

Page 4, Line 19. Should read '......concluding that THE modelled spatial distribution.....'.

Page 5, Line 3. What are the problems with the sensors that you refer to?

Page 5, Line 12. '....and are at larger distance from dust sources, and shorter distance to the ocean', does not make sense. How about '...and are further away from the dust sources, and closer to the ocean.'

Page 5, Line 25. 'rather too large in the model'. How about instead '.... are overestimated in the model output'.

Page 6, Line 4. Please explain where Storhofdi is in order to put the rest of the discussion in Section 3.1.2 into context.

Page 6, Section 3.1.2. I think you need to cite the work of Prospero et al. (2012) here.

Page 6, Line 25. Here you refer to 'sandy fields' for the first time. What do you mean with this term? Is this the same as 'sandy deserts', as referred to in the Introduction. Please define these terms.

Page 7, Line 1. Should read 'during THE winter season'.

Page 7, Line 12. Use of the word 'particular' is not right here.

Page 7, Line 14. 'Looking at total dust emissions from Iceland, 50% is emitted in 25 days, and 90% in 110 days of the year. Previous studies of long-term dust frequency reported 135 dust days per year (Dagsson-Waldhauserova et al., 2014).' Please expand on this, what conclusions do you draw, do you consider this to be a significant discrepancy, if so why is there a difference?

Page 7, Line 26. You refer to emission rates presented by Arnalds et al. (2014). Please provide details as to how these emission rates were determined.

Page 8, Line 12. 'To understand where dust that is emitted from Iceland can be found in the atmosphere and on the ground'. This sentence is a little clumsy. Could you describe this as 'to understand the transport of pathways of dust from Icleand..'?

Page 8, Line 29. Typo, remove '8'.

Page 9, Line 3. 'Baddock et al. (2017) did study trajectories from either south or north Iceland and showed that dust from south Iceland....'. Please improve this sentence. I would suggest: 'Baddock et al. (2017) studied the trajectories from sources in both the south and north of Iceland and showed that dust from south Iceland...'.

Page 9, Line 10. Please clarify what you mean here. You state that: 'A large fraction of emitted dust (<20 $\mu$m) does not travel far and is deposited in Iceland.' Do you mean

that you have found a large fraction of the emitted dust is on particles with diameter < 20 um? But I thought you only considered particles up to this diameter? Perhaps you are just reconfirming that you have only considered this size range?

Page 9, Line 26. 'especially varying' does not make sense. How about: 'deposition varied significantly'. Also, are you referring to deposition rates or where particles were deposited?

Page 10, Line 4. Please correct the sentence: 'In this study we made model simulations'. Incorrect use of the word 'made'.

Page 10, Line 14. Please correct the sentence: 'Best agreement with PM measurements over one year is found close to dust sources.' It does not make sense.

Page 10, Line 21. 'At Storhofdi, near the south coast of Iceland, the timing of peaks in dust concentrations is very well captured in our simulations, as we determined based on a comparison of modelled and measured dust concentrations between 1997 and 2002'. The structure of this sentence needs to be improved. Something along the lines of: '.......the timing of the peaks in dust concentration in our simulations compared well with the observed peaks in measured dust concentrations between 1997 and 2002'.

Page 10, Line 24. Please expand, which way does the dust from the north go?

Page 10, Lines 25 and 26. The use of the term 'much dust', is repetitive and clumsy.

Figures.

Figure 1. Please provide more details on how the soil fractions were determined, where does this data come from? How does soil fraction relate to 'dust' in this context? Is it possible to indicate where the 'dust hotspots' are. Please also improve the colour bar to indicate that 1.0 (?) is the maximum.

Figures 7 and 8. Please improve the labels on the colour bars. Figure 7b only has two! And neither 7a, 7b or 8a indicate the maximum value. Also, in my version there are no

labels for the individual figures (a and b).

---

## Author Comment (AC1) · 26 Jul 2017

RC: The authors present results from a muli-annual (27 years) study assessing the capability of the Lagrangian model FLEXPART to capture the Icelandic atmospheric dust life-cycle. Thereby, dust emission fluxes are estimated using FLEXDUST. Results of their study were further discussed regarding its interannual variability; results at high resolution were validated against measurements for the year 2012. The manuscript is well structured and a nice read. However, I do have some comments I would like the authors to address.

Authors: Thank you for your review.

2.1 Model description

(1) In the subsection FLEXDUST you describe how dust sources were implemented in the model. You state that lower friction velocities and large soil fractions were assigned to dust hot spots as identified by Arnalds et al. (2016). I am wondering whether these dust host spots occur due to enhanced levels of sediment supply or due to higher frequencies of stronger winds (maybe also channelled by orography).

Authors: The dust hot spots are known to be frequently active. Arnalds et al. (2016) ascribed this mostly to enhanced sediment supply, but also strong wind frequencies and soil properties (weaker winds can mobilize particles). Even without higher frequencies of stronger winds this already leads to larger dust emissions. To our knowledge no research has been published so far on strong wind frequency in dust hot spots.

(2) Can you spend some more words on how FLEXPART and FLEXDUST coexist respectively intertwine as this remains somewhat diffuse. As far as I understand FLEXDUST is used to estimate dust emission fluxes based on ECMWF forecast analyses at 0.2deg horizontal grid spacing. The calculated emission fluxes are then read into FLEXPART and transported whereby FLEXPART is driven using the ERA-Interim reanalysis at 1deg horizontal grid spacing. Why were two different atmospheric data sets chosen to drive the models rather than using consistently ECMWF forecast analyses for both but on a different horizontal grid?

Authors: Indeed FLEXPART and FLEXDUST are separate models. Our description of the simulation setup was obviously confusing. We always used the same ECMWF data for FLEXDUST and subsequent FLEXPART simulations. The high-resolution data were used for one year of model testing, whereas ERA-Interim data were used for the long-term simulations.
Changes: We changed the simulation descriptions in section 2.2 to clarify this.

(3) How is dust deposition respectively removal parameterized? Please add some explaining words. Is wash-out and scavenging due to rain and clouds considered as particle removal processes?

Authors: Yes, these processes are considered, as we mentioned in our manuscript: "In FLEXPART, simulated dust particles are influenced by gravitational settling, dry deposition and in-cloud and below-cloud scavenging (Grythe et al., 2016)." Deposition processes are described in detail by Grythe et al. (2017) and for interpretation of the current study it suffices to know that these processes were included, we therefore choose to give a reference rather than a description. However, we added one sentence to

explain a little better how removal processes are treated in FLEXPART: "Dry deposition is treated using the resistance method (Stohl et al., 2005), wet deposition distinguishes between liquid-phase and ice-phase scavenging (Grythe et al., 2016). "

(4) Simulation setup (section 2.2): As the input meteorological fields were available at a grid with a 0.2deg horizontal grid spacing, but dust emission fluxes were estimated on a grid with 0.01deg horizontal grid spacing, can you explain if there has been any upscaling or interpolation method applied, please? Is topography taken into account for the upscaling?

Authors: There was no upscaling involved for the meteorological fields, we use the 0.2 and 1.0 degrees grid values for the respective simulations. The surface type maps however, were available on a much higher resolution. Even though we use coarser-resolution wind fields, we can clearly define where dust emission occurs and this will give better initial conditions for Lagrangian modelling of particle trajectories. Notice that this method takes advantage of the Lagrangian nature of FLEXPART which is, in principle, independent of the resolution of the meteorological fields and thus can ingest emission data at any resolution.
Changes: We now comment on this in section 2.2.

3. Results and discussion
(5) In section 3.2.1, numbers of days of active dust emission are provided as fraction per annum. How do these numbers of days compare to seasons? Some additional sentences presenting and discussing the seasonal distribution of dust emission events, transport and deposition can help here to draw a more thorough picture of the Icelandic atmospheric dust life-cycle - and eventually imply further mechanism controlling interannual variability.
Authors: Modelled dust emission in Iceland is largest in winter/early-spring.
Changed: We added this to section 3.2.1.

(6) Is there any explanation why the NAO has no significant correlation with dust emission in Iceland?
Authors: It appears that the NAO index does not control dust storm frequency in Iceland. This was also concluded by Dagsson-Waldhauserova et al. (2014). Although we did not look at this in more detail, possible explanations may be found in increased precipitation or storm occurrence during seasonal snow cover.

(section 3.2.2)
(7) As stated in section 3.2.2, the NAO has no significant impact on dust emission. However, why is the NAO used as measure describing Aeolian transport and deposition patterns (section 3.3)? May topography has an important and maybe dominating impact on the transport direction here?
Authors: We hypothesised that even though emission is not linked to NAO, the transport patterns might be. For instance, pollution transport from Europe into the Arctic is strongly controlled by the NAO (Eckhardt et al., 2003). If dust would reach the south-east of Iceland where wind patterns (and thus transport patterns) correlate strongly with NAO, this might result in a correlation nonetheless. Even though no correlation was found, we think it is important to show this, as this was not clear a priori. Topography could be important as well as we also discuss in section 3.3 but we cannot explain this explicitly because we do not study transport pathways of specific regions.
Changes; We extended the discussion in section 3.2.2.

(8) How is the dust vertically distributed? Is there any significant dependency between dust deposition region and transport height or mixing depth into the boundary layer over source regions that can be concluded from the FLEXPART simulations? An enlarged discussion on dust transport pattern and deposition regions is desirable in order to clarify the conditions under which Icelandic dust is transported far beyond its source region. Furthermore, the results may vary with season as the predominance of meteorological situations (e.g. occurrence of precipitation, cloud formation) and atmospheric circulation patterns changes.

Authors: This is an interesting discussion, yet in our simulations we do not split dust from different source regions and we saved only limited data on the vertical distribution of dust. The modelled vertical distribution of Icelandic dust is limited. Global averages show that over 40% of suspended Icelandic dust is at altitudes less than 1000 m above the surface, thus probably within the atmospheric boundary layer. In averaged concentration fields only 6 % of suspended dust is situated at altitudes above 5000 m. Dust from the Hagavatn region has been observed at altitudes of 2 km and in LOAC (Renard et al., 2016a,b) vertical distributions dust reaches altitudes of 1 km during a dust-precipitation event in 2013 (not published).

Changes: We comment on the vertical distribution in section 3.3.

(9) Can the hypothesis by Meinander et al. (2016) that "Icelandic dust may have a comparable or even larger effect on the cryosphere than soot" be confirmed by the presented study?

Authors: This study confirms that Icelandic dust is likely to have an effect on the cryosphere and especially on the glaciers in Iceland, as can be concluded in combination with the results of Wittmann et al. (2017). However, this study was not set up to test this particular hypothesis and we would need to consider the complete cryosphere and include snowpack modelling influenced by soot as well as dust, and radiative transfer modelling. This may be a topic of further research.

---

## Author Comment (AC2) · 26 Jul 2017

This compact paper, "Temporal and spatial variability of Icelandic dust emission and atmospheric transport" presents surface observations and results of lagrangian simulations of dust emission and deposition at high resolution for 2012 and lower resolution for 1990-2016 to estimate the dust emission and deposition to the region. The paper is very well written, presents interesting results from modeling and the observations, and references the appropriate literature. I believe that details are lacking in places and the analysis is a little weak, namely the comparison with observations and the certainty with which the dust emissions can be estimated, and would like to see those parts improved prior to publication.

General comments

The PM10 and simulated dust concentration yield similar mean (21 ug/m3 and 28 ug/m3, respectively) and standard deviations (pg 5 line 10); however, this is comparing dust-only concentrations from the model with bulk aerosol PM10. This suggests that the simulated dust concentrations are actually biased high (maybe up to a factor 2?) relative to the observations (if the non-dust component could be removed from the PM10). I think the way that the model results are compared to the PM10 (and PM2.5) may need reconsidering or the present method better justified. Can you estimate the non-dust component? How much of the PM10 at the sites may be localized dust that would not be captured by the model? This affects the attempt to estimate the annual emissions from Iceland and subsequent deposition. I'm not sure whether the current observational constraints and analysis are able to fully support the estimate. The agreement with the dust concentration measurements seems reasonable at StórhöfÃˇri, but this is only a single measurement site SW of the source regions. Therefore, the constraint on emissions transported in other directions is weak; it appears that equal, if not great, dust mass is deposited to the NE. Could the statistical relationship between observations and modeled dust concentration be used to better estimate the emission, or at least the uncertainty? For example, how much would the emissions need scaling to provide the same average dust concentration (or some other metric) at StórhöfÃˇri? this suffers from the lack of constraints for the dust in the NE, but might give a better representation of the dust emissions and their uncertainty beyond the interannual variability.

Authors: Thank you for your review. Indeed the constraints on dust emission in Iceland are weak. This results from the paucity of available data in Iceland and, to our knowledge, we have used all data that are available to compare our simulations with, even if most of the measurements (especially the PM measurements) do not allow direct comparisons. In the paper, we simply tried to use the long-term measurements that are currently available and also suggest that future more specific measurements would be needed to quantify the apparently important Icelandic dust sources. PM10 and PM2.5 values include other aerosols. This is of less concern at Raufarfell where traffic and sea salt influence are considered limited, but of larger concern in domestic and coastal areas as discussed in section 3.1.1. We therefore give more emphasis to the measurements at Storhofdi that only include dust. Close to the dust sources in NE Iceland we could confirm deposition rates based on snow sample observations (Wittmann et al., 2017). Other quantitative data is unfortunately not available and there are also no other supportive data that would allow a speciation of the PM data into different aerosol types. With the current data we can only conclude that timing of dust events can be captured and that dust deposition and concentrations, and therefore expectedly dust emission, are on the right order of magnitude. More precise estimates on these scales are currently not feasible, yet the model does provide an upper constraint.

Changes: we added and discuss references on different sources causing PM10 values exceeding health limits in Reykjavik and aerosol concentrations (other than dust) at Storhofdi in section 3.1.1.

Following from this, I can't see any comparison of the low and high resolution runs in 2012 (other than 2.9 Tg and 5.1 Tg totals for 2012 on pg7, line 21). Does this mean that running at high resolution may give 75% higher emission estimates than the 4.3+/- 0.8 Tg presented for the long term estimates? I don't

think the implications of this are discussed clearly enough. The uncertainty estimate for the interannual variability may mislead the reader to the certainty of the magnitude of the dust emissions (and hence deposition). Does the low resolution run well-reproduce the high resolution simulated dust concentration timing in 2012 otherwise? Maybe add the low resolution timeseries at StórhöfÃˇri to Figure 2?

Authors: In 2012 emission estimates were higher based on high resolution data. Deviations are also likely for other years. The measurements at Storhofdi are not available in 2012, instead we now discuss the high and low resolution runs in comparison to the PM10 measurements.

Changes: we provided additional results from the low-resolution simulation and extended the discussion on the influence of resolution in section 3.1.1.

While the time series of concentration provide a good visual reference of the frequency and magnitude of events, they are not ideal for illustrating the agreement between the simulation and the observations. I recommend providing a scatter-plot (perhaps on a log-log scale?) to better illustrate how well the model captures the observations of dust concentration. This is less useful (and therefore perhaps less necessary) for the comparison with PM for the reason outlined above, unless speciation is available.

Authors: We agree that this is useful for the dust concentrations, in combination with the time series already given.

Changes: We added such a figure for the Storhofdi data.

Emissions are not allowed when the precipitation rate is above the 1 mm/hr threshold. Is there a lag time for this emission suppression after the rain stops? Or is it expected that the timescale for the surface drying and becoming an active once again is shorter than the model timestep? How much do you think this assumption affects the emission?

Authors: In FLEXDUST, there is no time lag suppressing dust emission after rain. In a model test case where dust emission is prohibited if the precipitation sum in the past 4 hours exceeds 2 mm the model failed to simulate some strong dust events that were recorded in PM10 measurements. We therefore assume that during strong wind conditions the top sediment layer is quickly dried and dust emission is possible. This assumption is confirmed by several observations. Dagsson-Waldhauserova et al. (2014b) observed dust mobilization of wet particles, even during low-wind conditions. They discussed that the relatively dark basaltic dust might dry quickly. Also during intermittent snowfall dust mobilization has been recorded (Dagsson-Waldhauserova et al., 2015). Furthermore, analysis of long-term weather observations of dust events (e.g. Dagsson-Waldhauserova et al., 2014a) revealed that suspended dust is observed during precipitation events, although it is noted that precipitation at the weather observation location does not necessarily imply wet conditions at the dust source.

Changes: We added a model test for PM10 concentrations at Raufarfell and discuss the model results and references in section 3.1.

In Groot Zwaaftink et al. (2016) it is stated that, relative to a precipitation threshold, "Especially in northern latitudes, soil moisture appeared a better indicator of mobilization threshold as seasonal variations in surface dust concentrations at remote stations were better captured and total emission amounts were closer to other model estimates." Please can you comment on why this is different to the current research findings for Iceland.

Authors: The global simulations were based on a combination of size-dependent friction velocity thresholds (Shao and Lu, 2000) and increase of threshold friction velocity due to soil moisture according to Fécan et al. (1999). Instead, we here use friction velocities from field observations in Iceland. The combination of these observed thresholds with the soil moisture parameterization of Fécan et al. (1999) lead to very low dust emission rates and dust concentrations far below the measured values shown in section 3.1, as also discussed in our manuscript. Dagsson-Waldhauserova et al. (2014b) noted that the relatively dark basaltic dust of an Icelandic dust source dried quickly and dust mobilization occurred during moist conditions. It could thus be that Icelandic dust mobilization is less dependent on soil moisture and the soil moisture parameterization (Fécan et al., 1999) is not applicable. Furthermore, the ECMWF soil moisture data might not be representative for the layer from which dust is mobilized.

Changes; we extended the discussion on the soil moisture parameterization in section 2.1.

Specific Comments
pg 3 line 18 - "FLEPXART" typo
Authors: Corrected
pg 7 line 19 "FLEXUDST" typo
Authors: Corrected

It may be clearer to refer to the "soil fraction" as the "bare soil fraction" throughout.
Authors: Yes, that's better, we changed this.

Table 1 - it isn't quite clear how these values are derived from the threshold friction velocities presented in Arnalds et al. (2001) and the discussion in Arnalds et al. (2016). Please can you elaborate in the text on how these values are derived.
Authors: We added the explanation below in section 2.1.
Changes: We use observations from Arnalds et al. (2001) and a description of erosion levels (Arnalds et al., 2016) to determine the threshold friction velocity (see Table 1). While Leadbetter et al. (2012) and Liu et al. (2014) chose a fixed threshold friction velocity of 0.4 m s-1 for mobilization of volcanic ash, the range of values applied here is more suitable to cover the different conditions of multiple dust sources. Arnalds et al. (2016) give an overview of erosion classes for each surface type. For regions with extremely severe erosion we assume the average of threshold values observed at several sand fields, for severe erosion we assume average conditions of sandy gravel and for considerable erosion we apply an upper threshold observed for sandy gravel (Arnalds et al., 2001). So called dust hot spots, described by Arnalds et al. (2016), were also included in our simulations. These were assigned a lower friction velocity (see Table 1), corresponding to the lowest threshold wind velocity estimates for erosion by Arnalds et al. (2016), and a slightly larger bare soil fraction (+3%).

Figure 2 - The Raufarfell timeseries is hard to see because of the upper limit. Is it possible to use a discontinuity on the y-axis above _600 ug/m3 to better visualize the data at lower concentrations.
Changes: We changed the figure accordingly.

---

## Author Comment (AC3) · 26 Jul 2017

The paper presents a modelling study of the emission and transport of dust in Iceland between 1990 and 2016. It highlights the significance of high latitude dust sources on the global dust budget, and the authors present interesting results showing the main transport pathways of dust from Iceland. However, I believe the description of the model set-up needs to be significantly improved before this paper can be published. Details, including a description of the resolution of the model topography used and the particle size distribution applied are missing, and there needs to be some discussion on how their results may be sensitive to their set-up. The manuscript would also be improved by including some discussion on how the supply of new dust sources, related to volcanic eruptions in Iceland, might influence their results.

Authors; Thank you for your constructive review.

1. The Introduction
I can see the importance and relevance of this study but I don't think this is reflected in the introduction. Details are missing and statements are often not backed up with existing data and/or references are missing. Currently, it reads as a series of statements rather than explaining to the reader why the study is important, the approach, and how it fits in with the existing literature. You need to discuss in more detail the work that has previously been carried out to better understand dust emissions in Iceland, including work published by Olafur Arnalds and Pavla Dagsson-Waldhauserova, and you should consider work on dust events in other parts of the world too. Further discussion on modelling dust emissions is also needed. You state that model simulations of dust emissions in Iceland are lacking but there is now a body of work on modelling remobilisation of volcanic ash in Iceland, see Leadbetter et al. (2012), Liu et al. (2014), Beckett et al. (2017), and further afield, for example Folch et al. (2013) and Mingari et al. (2017) who consider remobilization in Argentina. Given that volcanic ash is a significant source of PM in Iceland (indeed there is the question of what is dust and what is ash!!), and the modelling approaches for remobilized ash are very similar to the approach you have applied here you should discuss this.

Authors; Indeed the modelling efforts considering remobilisation of volcanic ash are relevant and are now included in the introduction. We also added more details on current knowledge of Icelandic dust, although we refer to a recent review paper by Arnalds et al. (2016) for a complete overview.

Specific comments:
Line 3: You state that: 'Model simulations indicated that 0.3% of global dust emission may originate from Iceland (Groot Zwaaftink et al., 2016)'. More details are needed here, what model, what simulations were performed and with what aim? If this has already been done then where does the study you are about to present fit in?

Authors; These were global simulations over a three-years periods where spatial distribution of dust emission in Iceland was not discussed.
Changes; We give additional details on this reference.

Line 4: You state that 'it is known that dust storms frequently occur there [Iceland]' and cite Dagsson-Waldhauserova et al. (2014). It would be good to include some numbers here e.g. how many dusty days, on average, occur in Iceland. This will help put your results into context later on too. I realise you comment on this later in the paper but this should be here in the Introduction.

Authors; We added this information in the introduction.

Line 14: You need to provide a reference for the surface type map that you refer to.
Authors: Added.

2. Model Set Up
The explanation of your model set-up is missing many details. I think you should include the equations used in FLEXDUST to model the emission of dust, and explain the variables. Exactly how does your model set-up account for topography, snow cover and soil moisture? You state that precipitation halts mobilization. You need to refer to the work of Leadbetter et al. (2012) here who also considered how best to represent the impact of precipitation on mobilization of volcanic ash in Iceland. Please can you also comment on how well you think this approach is working in respect to representing the timing and frequency of dust events? This is discussed by Leadbetter et al. (2012) and Liu et al. (2014) who both point out that this approach does not account for wetting and drying of volcanic ash deposits, do you think this is true of all dust sources?
Authors; FLEXDUST equations have been given in Groot Zwaaftink et al. (2016) and we do not think this should be repeated here. We rather concentrated our presentation of FLEXDUST on the differences in the model set-up used in the present paper from the one used by Groot Zwaaftink et al. (2016), although we agree that a little more detail will be helpful for the reader. Indeed precipitation influences dust mobilization, as is accounted for in the model and we agree that more discussion on this topic is useful.
Changes: We added equation 1, which gives the dependency of dust emission on (threshold) friction velocity. We now refer to these studies on mobilization of volcanic ash in sections 1 and 2. We also added results of a test simulation where we included a drying period after precipitation that showed dust mobilization was not better represented near the source by inclusion of such a time lag. We also slightly extended the general description of FLEXDUST.

Please provide the Particle Size Distribution (PSD) you used and explain your reasoning for this choice. Why did you choose to consider particles with diameter up to 20 um only? What is the minimum particle size you considered? The work of Liu et al. (2014) gives the PSD of ash particles that had been remobilized and deposited in Reykjavik during March 2013. They found that particles had a mode at 32-63 um. Have there been any measurements of the PSD of particles mobilised from the other dust sources in Iceland?
Authors; We considered particles in the size range 0.2 to 20 µm, consistent with global dust simulations using FLEXDUST and FLEXPART (Groot Zwaaftink et al., 2016). The size distribution is provided in the given references. It is quite standard to consider only particle sizes less than20 µm in dust modelling (e.g. Tegen, 2003). Observations of size distributions in Icelandic dust storms show that particle mean diameter is much smaller than 10 µm (Dagsson-Waldhauserova et al., 2014b). Larger particles may be present close to the sources but their potential for atmospheric transport away from the source region is very limited, due to rapid gravitational settling. As our focus is on dust transport, we do not include such large particles in our simulations.
Changes: we added the minimum particle size to section 2 and discuss observed particle size distributions.

3. Thresholds Friction Velocities.
Please provide your reasoning for the threshold friction velocities that you apply. How were these values determined from the Arnalds et al. (2001) and Arnalds et al. (2016) papers and how are the classes defined? Please also provide information on how these classes are distributed across Iceland. Figure 1 shows the soil fractions applied but please also highlight where the Dust Hot Spots are and how the erosion classes are applied across the other source regions. Please can you also comment on how good a job you think these threshold friction velocities are doing. By applying this range of values are you doing a good job of representing the timing and frequency of events in your model output? How sensitive is your model output to the threshold friction velocity applied? Can you account for some of the mismatch between the observed and modelled PM10 and PM2.5 air concentrations if you vary the threshold friction velocity applied?

Authors; The erosion classes have been presented in several publications (Arnalds et al., 2001; 2014; 2016) and are therefore not repeated here. The dust hot spots are the regions with maximum soil fraction in Figure 1. The threshold friction velocity affects timing and frequency of dust events and the concentration during events, as is now also clear from equation 1. Some of the mismatches are likely related to threshold friction velocity, we expect mostly because we use a fixed threshold (besides precipitation and snow cover influence). The threshold friction velocities of several sources are probably changing over time, even during dust events the surface conditions are changing. Despite the strong simplifications we apply in our model, we are able to capture timing of several events.

Changes; We added equation 1, which shows dependency of dust emission on threshold friction velocity. We added an explanation on the thresholds for each erosion class and a discussion in section 2.1. We discuss the influence of threshold friction velocity on results in sections 3.1.1, 3.1.2 and 4.

4. Topography

What is the resolution of your model topography? Are your results sensitive this? You state in the introduction that dust events can be driven by katabatic winds; does your model topography allow you to capture these meteorological phenomena? Mingari et al. (2017) show how the topography in Argentina influences the local winds and in turn how that drives mobilization. I think you need to consider this. This information would help put in context your later comment in Section 3.1.1 that the model output may not be able to capture observed PM10 concentrations because of the resolution of the topography.

Authors; The topography resolution is the same as in the ECMWF wind fields, thus 0.2 degrees for the high-resolution simulation and 1 degree for the long-term simulations. Indeed we cannot capture all local winds and discuss this in our manuscript.

Changes: We now already introduce this potential problem in section 2. We also add a discussion on sensitivity to model resolution in section 3.1.1.

5. Sources

You compare your model output air concentrations to PM data from monitoring stations across Iceland collected during 2012. You state that: 'In this year no volcanic eruptions occurred that could strongly influence PM measurements' (Line 8, Section 2.3). I disagree. Do you really think that the ash deposits from the eruption of Grimsvotn only the year before and from Eyjafjallajökull in 2010 had all been removed and were no longer a significant source of PM? The study by Leadbetter et al. (2012) considers the remobilization of volcanic ash from the deposits resulting from the eruption of Eyjafjallajökull in 2010. They compared modelled air concentrations using the dispersion model NAME, which includes a resuspension scheme, to PM10 measurements across Iceland during September 2010 to February 2011. Their modelled concentrations agree well with the timing and location of observed peaks in the PM10 data from the monitoring stations, and here only the Eyjafjallajökull ash is defined as the source. I recognize that your study aims to consider the long-range trends of dust emission and transport from sources across Iceland, but I think you need to acknowledge the fact that volcanic eruptions result in significant new sources of unconsolidated deposits which can continue to be remobilized for years after an eruption. In Section 3.2.2 you go on to state that your modelled dust emission rates are an order of magnitude lower than previous estimates given by Arnalds et al. (2014), and you say this could be related to volcanic events. I would suggest that you could explore this further and consider that the deposits from the Grimsvotn and Eyjafjallajökull eruptions could be a significant source of PM in your study.

Authors; We mainly wanted to avoid influence from direct injection of volcanic ash into the atmosphere. Resuspension of deposited tephra should be included in FLEXDUST, so in principle does not constitute a problem. Iceland, generally, is highly dynamic and land cover changes in response to volcanic eruptions and as deposited tephra fields age. In 2012, there was no volcanic eruption in Iceland, but of course ash deposits from previous years may still be remobilized. In fact, the dust sources in our surface type map are partly covered with fresh tephra. Also ash from the Eyjafjallajökull and Grimsvotn eruptions were partly deposited on active dust sources that are included in our model, even though our land cover map does not account for any changes due to the recent eruptions. This means that we partly include resuspension of volcanic material. This should indeed be included as a discussion and we added this in sections 2 and 4.

6. The impact of NAO

I did not follow why you chose to consider the role of NAO as part of your study and what the significance is? What meteorological variables and/or synoptic conditions related to NAO do you think impact mobilization events in Iceland?

Authors; The winter Icelandic low is stronger during NAO positive phases according to model simulations (Bromwich et al., 2005) and this relates to precipitation, temperature and wind in Iceland. Stronger winds can enhance dust mobilization, while precipitation and snow cover can inhibit dust mobilization. We thus wanted to know if dust emission amounts are related to NAO. Furthermore, stronger winds over the North Atlantic can increase dust transport.

Changes: We changed the discussion on NAO in section 3.3.

Minor Comments

In several places, including in the abstract, you state your conclusion that: 'Annual dust emission amounts to 4.3_0.8 Tg during the 27 years of simulation'. I find the term 'amounts to' a little confusing when discussing the yearly average. Please clarify.

Changes; we rephrased where applicable.

Page 1, Line 3. Emission should read emissions.

Authors: rephrased

Page 1, Line 19. 'A model for estimates of dust emission', does not read very well. The structure of this sentence needs to be improved.

Changes; rephrased.

Page 1, Line 26. Please provide references for your examples on the impacts of dust.

Authors; we provide references later in the introduction.

Page 2, Line 14. '.......surface type map of Iceland to identify dust sources'. I think you need to cite Arnalds et al. (2016) here.

Changes; we added an appropriate reference.

Page 2, Line 21. I did not quite follow this sentence: ' ....and originally accounts for snow cover, topography....' . What do you mean by 'originally'?

Authors: It does in the global setup where the model was first introduced, but this differs in the Iceland version.

Changes; rephrased

Page 3, Line 3. 'As we here mainly deal with sediments'. What do you mean by this statement, what is the relevance of 'sediments' is this different to 'dust'. Please clarify. Also the structure of this sentence could be better, what do you mean by 'mainly deal with'?

Changes; Rephrased this section.

Page 3, Line 8. What do you mean by a 'closed snow cover'?

Authors; A snow cover that does not consist of snow patches but covers the area.

Page 3, Line 17. 'as was previously also done for'. Could read better, how about 'and has previously been used to model the transport of Saharan dust'.

Changes: rephrased

Page 3, Line 21. What do you mean by a 'multitude of particles'? Please be specific.

Changes: rephrased

Page 4, Lines 1 and 2. Here you write the units of the particle size (micrometre), in other places you use the symbol. Please correct. Also, the structure of this sentence could be improved.

Changes: rephrased

Page 4, Line 17. 'Model evaluation is limited due to a lack of data.' This sentence does not read well. Please improve the structure of this paragraph.
Changes: rephrased

Page 4, Line 19. Should read '......concluding that THE modelled spatial distribution.....'.
Changes: rephrased

Page 5, Line 3. What are the problems with the sensors that you refer to?
Authors; there were different problems, but further details will not improve understanding of the results.

Page 5, Line 12. '....and are at larger distance from dust sources, and shorter distance to the ocean', does not make sense. How about '...and are further away from the dust sources, and closer to the ocean.'
Changes; rephrased.

Page 5, Line 25. 'rather too large in the model'. How about instead '.... are overestimated in the model output'.
Changes: rephrased

Page 6, Line 4. Please explain where Storhofdi is in order to put the rest of the discussion in Section 3.1.2 into context.
Changes: rephrased

Page 6, Section 3.1.2. I think you need to cite the work of Prospero et al. (2012) here.
Changes: rephrased

Page 6, Line 25. Here you refer to 'sandy fields' for the first time. What do you mean with this term? Is this the same as 'sandy deserts', as referred to in the Introduction. Please define these terms.
Changes: rephrased

Page 7, Line 1. Should read 'during THE winter season'.
Changes: rephrased

Page 7, Line 12. Use of the word 'particular' is not right here.
Changes: rephrased

Page 7, Line 14. 'Looking at total dust emissions from Iceland, 50% is emitted in 25 days, and 90% in 110 days of the year. Previous studies of long-term dust frequency reported 135 dust days per year (Dagsson-Waldhauserova et al., 2014).' Please expand on this, what conclusions do you draw, do you consider this to be a significant discrepancy, if so why is there a difference?
Changes: we find this a good agreement and now comment on this in the manuscript.

Page 7, Line 26. You refer to emission rates presented by Arnalds et al. (2014). Please provide details as to how these emission rates were determined.
Changes; we added details in the introduction and rephrased this section.

Page 8, Line 12. 'To understand where dust that is emitted from Iceland can be found in the atmosphere and on the ground'. This sentence is a little clumsy. Could you describe this as 'to understand the transport of pathways of dust from Iceland..'?
Changes: rephrased

Page 8, Line 29. Typo, remove '8'.
Changes: removed

Page 9, Line 3. 'Baddock et al. (2017) did study trajectories from either south or north Iceland and showed that dust from south Iceland....'. Please improve this sentence. I would suggest: 'Baddock et al. (2017) studied the trajectories from sources in both the south and north of Iceland and showed that dust from south Iceland...'.
Changes: rephrased

Page 9, Line 10. Please clarify what you mean here. You state that: 'A large fraction of emitted dust (<20 _m) does not travel far and is deposited in Iceland.' Do you mean that you have found a large fraction of the emitted dust is on particles with diameter <20 um? But I thought you only considered particles up to this diameter? Perhaps you are just reconfirming that you have only considered this size range?
Authors; indeed we wanted to clarify that we only consider dust <20 um.
Changes; we removed (<20 _m)

Page 9, Line 26. 'especially varying' does not make sense. How about: 'deposition varied significantly'. Also, are you referring to deposition rates or where particles were deposited?
Changes: rephrased

Page 10, Line 4. Please correct the sentence: 'In this study we made model simulations'. Incorrect use of the word 'made'.
Changes: rephrased

Page 10, Line 14. Please correct the sentence: 'Best agreement with PM measurements over one year is found close to dust sources.' It does not make sense.
Changes: rephrased

Page 10, Line 21. 'At Storhofdi, near the south coast of Iceland, the timing of peaks in dust concentrations is very well captured in our simulations, as we determined based on a comparison of modelled and measured dust concentrations between 1997 and 2002'. The structure of this sentence needs to be improved. Something along the lines of: '.......the timing of the peaks in dust concentration in our simulations compared well with the observed peaks in measured dust concentrations between 1997 and 2002'.
Changes: rephrased

Page 10, Line 24. Please expand, which way does the dust from the north go?
Changes: rephrased

Page 10, Lines 25 and 26. The use of the term 'much dust', is repetitive and clumsy.
Changes: rephrased

Figures.
Figure 1. Please provide more details on how the soil fractions were determined, where does this data come from? How does soil fraction relate to 'dust' in this context? Is it possible to indicate where the 'dust hotspots' are. Please also improve the colour bar to indicate that 1.0 (?) is the maximum.
Authors; we assigned soil fractions to surface types, the dust hot spots are the locations with maximum bare soil fraction.
Changes; we changed the figure and add a comment in section 2.1

Figures 7 and 8. Please improve the labels on the colour bars. Figure 7b only has two!
And neither 7a, 7b or 8a indicate the maximum value. Also, in my version there are no labels for the individual figures (a and b).
Changes; changed

---

## Referee Report (RR1)

Review of revised Manuscript: 'Temporal and spatial variability of Icelandic dust emission and atmospheric transport' by Groot Zwaaftink et al.

By Frances Beckett

The authors have addressed all of my concerns in the revised version of the manuscript. The paper presents an interesting study which represents a valuable addition to the subject area, and the introduction now better reflects this. The additional discussion on the particle size distribution (PSD) of resuspended material is appreciated, and I read with interest the section which now considers a time lag on remobilization after precipitation. The additional discussion considering the sensitivity of the results to the model resolution, I think, adds greatly to the work. Finally, I thank the authors for the consideration of volcanic ash sources in this manuscript. I hope that this will lead to better understanding of remobilization of both volcanic ash and dust sources in Iceland, work on which has to date remained disparate. I agree that the volcanic ash and dust sources are closely related and should be unified.